# Cryo-EM study of an archaeal 30S initiation complex gives insights into evolution of translation initiation

Pierre-Damien Coureux [1]*, Christine Lazennec-Schurdevin[1], Sophie Bourcier[2], Yves Mechulam [1] & Emmanuelle Schmitt [1]*

Archaeal translation initiation occurs within a macromolecular complex containing the small ribosomal subunit (30S) bound to mRNA, initiation factors aIF1, aIF1A and the ternary complex aIF2:GDPNP:Met-tRNA$_i^{Met}$. Here, we determine the cryo-EM structure of a 30S:mRNA:aIF1A:aIF2:GTP:Met-tRNA$_i^{Met}$ complex from *Pyrococcus abyssi* at 3.2 Å resolution. It highlights archaeal features in ribosomal proteins and rRNA modifications. We find an aS21 protein, at the location of eS21 in eukaryotic ribosomes. Moreover, we identify an N-terminal extension of archaeal eL41 contacting the P site. We characterize 34 $N^4$-acetylcytidines distributed throughout 16S rRNA, likely contributing to hyperthermostability. Without aIF1, the 30S head is stabilized and initiator tRNA is tightly bound to the P site. A network of interactions involving tRNA, mRNA, rRNA modified nucleotides and C-terminal tails of uS9, uS13 and uS19 is observed. Universal features and domain-specific idiosyncrasies of translation initiation are discussed in light of ribosomal structures from representatives of each domain of life.

[1] Laboratoire de Biologie Structurale de la Cellule, BIOC, Ecole polytechnique, CNRS, Institut Polytechnique de Paris, 91128 Palaiseau, cedex, France. [2] Laboratoire de Chimie Moléculaire, LCM, Ecole polytechnique, CNRS, Institut Polytechnique de Paris, 91128 Palaiseau, cedex, France. *email: pierre-damien.coureux@polytechnique.edu; emmanuelle.schmitt@polytechnique.edu

Translation initiation universally occurs with accurate selection of the start codon that defines the reading frame on the mRNA. The mechanism involves a macromolecular complex composed of the small ribosomal subunit, the mRNA, a specialized methionylated initiator tRNA and initiation factors (IFs). Even if the process is universal, the molecular mechanisms are different in the three domains of life. In bacteria, the ribosome generally binds in the vicinity of the AUG start codon through interaction of 16S rRNA with the Shine-Dalgarno (SD) sequence on mRNA. The initiator tRNA is formylated and only three IFs, IF1–IF3 assist the start codon selection mechanism[1]. In eukaryotes, translation initiation is more complex with maturated mRNAs and many IFs. The canonical mechanism involves a pre-initiation complex (PIC) comprising the ternary complex eIF2:GTP:Met-tRNA$_i^{Met}$ (TC), the two small IFs eIF1 and eIF1A, as well as eIF5 and eIF3. In the presence of factors belonging to the eIF4 family, the PIC is recruited near the 5′-capped end and scans the mRNA until an AUG codon in a correct context (Kozak motif) is found. AUG recognition stops scanning, provokes factor release and the assembly of an elongation-proficient 80S complex through large subunit joining, with the help of eIF5B and eIF1A. eIF1 and eIF1A are key factors of the scanning process favoring a $P_{OUT}$ conformation of the TC, where tRNA is not fully accommodated in the P site. The multimeric factor eIF3 also stimulates attachment of the PIC to the mRNA and scanning. Finally, eIF5 is the guanine activating protein of eIF2 that stimulates GTP hydrolysis during scanning of the mRNA[2–4].

In archaea, genomic analyses have shown that three IFs homologous to their eukaryotic counterparts, aIF1, aIF1A, and aIF2 are found[5,6]. Moreover, archaeal ribosomal proteins are either universal or shared with eukaryotes showing the proximity of the two ribosomes[7–9]. However, there is no long-range scanning because mRNAs have SD sequences or very short 5′ untranslated regions (UTR) that allow pre-positioning of the IC in the vicinity of the start codon. Despite this, a local search of the mRNA by the IC (termed as "local scanning" in ref. [10]) is necessary to allow precise positioning of the start codon in the P site. Thus, even if the recruitment of the PIC on the mRNA is different in eukaryotes and archaea, start codon selection is achieved within a common structural core made up of the small ribosomal subunit, the mRNA, the methionylated initiator tRNA (Met-tRNA$_i^{Met}$) and the three IFs e/aIF1, e/aIF1A and e/aIF2[10]. e/aIF2 is a specific eukaryotic and archaeal heterotrimeric protein that binds the initiator tRNA in the presence of GTP[11]. Previous biochemical and structural studies of a full IC from *Pyrococcus abyssi* (30S:mRNA:TC:aIF1:aIF1A) identified two conformations with the initiator tRNA either in a remote position (IC0-P$_{REMOTE}$) or bound to the P site (IC1-P$_{IN}$). This led us to propose that conformational changes of the TC may participate in start codon selection[12,13]. In our model (named "spring force model" in ref. [11]), interaction of aIF2 with h44 of the 30S would counteract accommodation of the tRNA in the P site. However, formation of correct codon–anticodon pairing in the P site would compensate for the restoring force exerted by aIF2 on the tRNA. This would allow a longer stay of the initiator tRNA in the P site and trigger further events, including aIF1 departure because of steric hindrance and release of aIF2 in its GDP bound form. The role of aIF1-induced dynamics of the IC in the start codon selection was supported by toeprinting experiments[14]. In the absence of aIF1, the IC becomes more stable, as observed by a restricted toeprinting signal. However, no structural view of an archaeal IC illustrating a state following aIF1 departure has been described to date.

In the present study, we determine the cryo-EM structure of an archaeal IC (IC2, 30S:mRNA:aIF1A:aIF2:GTP:Met-tRNA$_i^{Met}$) from *P. abyssi* devoid of aIF1 at an overall resolution of 3.2 Å. Full reconstruction of an atomic model of the small ribosomal subunit highlights archaeal features in ribosomal proteins and rRNA modifications. We find a previously unidentified archaeal ribosomal protein aS21, at the location of eS21 in eukaryotic ribosomes. Moreover, a previously unobserved N-terminal extension of eL41 contacts the P site. We also identify a set of 34 N$^4$-acetylcytidines distributed throughout the 16S rRNA. These base modifications likely participate in the hyperthermostability of this ribosome. In the absence of aIF1, the 30S head is no longer mobile and the initiator tRNA becomes stably bound to the P site. A network of interactions involving rRNA modified nucleotides and the C-terminal tails of three universal ribosomal proteins, uS9, uS13, and uS19 is observed. Universal features and domain-specific idiosyncrasies of translation initiation are discussed in light of ribosomal structures from representatives of each domain of life.

## Results

**Overview of the IC2 cryo-EM structure**. In order to study the impact of aIF1 departure during translation initiation, we prepared an initiation complex (IC2) without this factor. IC2 contains archaeal 30S subunits from *P. abyssi* (Pa-30S), Pa-aIF1A, the ternary complex (Pa-aIF2:GDPNP:Met-tRNA$_i^{Met}$A$_1$-U$_{72}$) and a synthetic 26 nucleotide-long mRNA. Cryo-EM images were collected on a Titan Krios microscope (Table 1). After image processing, 218 k particles were used for refinement without classification. A density map to 3.2 Å resolution was obtained, showing a very good structural homogeneity of the complex. Density subtractions of the head or the body parts of the 30S further improved map quality (Supplementary Fig. 1). The high resolution of the electron density map allowed complete reconstruction of the 30S, as described below. After density subtraction and classification in RELION[15], one class showed very weak electron density for aIF1A and was therefore not further refined (IC2C, see Methods). The other classes showed two conformations of the initiation complex, named IC2A (34k particles, 4.2 Å resolution, Fig. 1a) and IC2B (142k particles, 3.3 Å resolution, Fig. 1b). The corresponding models were refined in PHENIX[16] (Supplementary Figs. 1–3, Table 1 and Supplementary Table 1). In the two conformations, the initiator tRNA and the mRNA are firmly bound to the 30S. Moreover, the position of the 30S head does not detectably change as compared to IC1-P$_{IN}$[12]. As already observed for ribosomal initiation complexes, local resolution of the electron density is higher for the ribosome core than for the peripheral IFs (Supplementary Fig. 3). As a consequence, aIF2-subunits and aIF1A were placed in the density by rigid body fitting of crystallographic structures (Table 1 and Methods). The two mobile wings of aIF2 (aIF2β core domain and aIF2α domain 1–2) are poorly defined in the two conformations and their positions were only tentatively modeled. As observed in IC0 and IC1[12], aIF1A is located in the A site. Its position corresponds to that observed for eukaryotic eIF1A in PIC[17–19]. aIF2 is bound to the 3′ aminoacylated end of the initiator tRNA. However, whereas domain III of aIF2γ (aIF2γDIII) contacts h44 in IC2A, the contact is lost in IC2B. Departure of aIF2γ from h44 is accompanied by a local movement of the rRNA helix (Supplementary Fig. 4). Because the non-hydrolyzable GTP analog, GDPNP, has been used in complex preparation, aIF2 is not fully released (Fig. 1).

**Identification of archaeal specificities of Pa-30S**. Complete building of the Pa-30S subunit was performed in the 3.2 Å resolution cryo-EM map using the medium resolution archaeal 30S structures as guides[12,20] (Fig. 2a). Statistics of the refined structure and final secondary structure diagram of Pa-16S rRNA

**Table 1 Cryo-EM statistics.**

| Data collection and processing | IC2 body EMDB-10323 PDB 6SWD | IC2 head EMDB-10324 PDB 6SWE | IC2A EMDB-10320 PDB 6SW9 | IC2B EMDB-10322 PDB 6SWC |
|---|---|---|---|---|
| Magnification | | ×128,440 | | |
| Voltage (kV) | | 300 | | |
| Electron exposure (e⁻/Å²) | | 32 | | |
| Defocus range (μm) | | −0.5 to −5 | | |
| Pixel size (Å) | | 1.09 | | |
| Symmetry imposed | | C1 | | |
| Initial particle images (no.) | | ~332,000 | | |
| Final particle images (no.) | ~218,000 | ~218,000 | ~34,000 | ~142,000 |
| Resolution (unmasked, Å) | 3.2 | 3.1 | 4.8 | 3.5 |
| Resolution (masked, Å) | 3.2 | 3.1 | 4.2 | 3.3 |
| Initial models used (PDB codes) | 4V6U, 5JB3 | | 4V6U, 5JB3, 4RD4, 2QMU, 4MNO | |
| Model resolution (Å) | 3.2 | 3.1 | 4.2 | 3.3 |
| Map sharpening B factor (Å²) | −86 | −88 | −107 | −92 |
| Model composition | | | | |
| Nonhydrogen atoms | 41,559 | 22,797 | 70,662 | 70,737 |
| Protein residues/nucleotides | 2293/1039 | 1525/479 | 4508/1560 | 4518/1560 |
| Ligands | 55 | 17 | 74 | 74 |
| B factors (Å²) | | | | |
| Protein | 26 | 52 | 135 | 57 |
| Nucleic acid | 38 | 44 | 106 | 52 |
| Ligand | 22 | 35 | 77 | 33 |
| R.m.s. deviations | | | | |
| Bond lengths (Å) | 0.012 | 0.010 | 0.011 | 0.010 |
| Bond angles (°) | 1.012 | 0.937 | 1.207 | 0.895 |
| Validation | | | | |
| MolProbity score | 1.91 | 2.00 | 2.68 | 2.13 |
| Clashscore | 5.65 | 7.31 | 29.24 | 9.42 |
| Poor rotamers (%) | 1.12 | 0.55 | 0.80 | 0.05 |
| Ramachandran plot | | | | |
| Favored (%) | 89.29 | 88.04 | 80.8 | 86.62 |
| Allowed (%) | 10.62 | 11.96 | 19.04 | 13.35 |
| Disallowed (%) | 0.09 | 0 | 0 | 0.02 |

Data collection, processing, and refinement statistics for IC2A, IC2B, IC2 body, and IC2 head datasets.

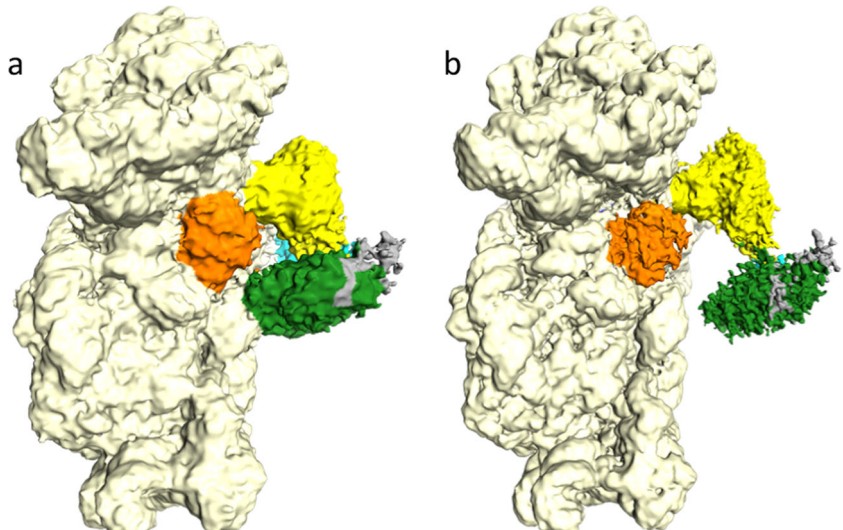

**Fig. 1 Cryo-EM maps of two conformations of the initiation complex 30S:mRNA:aIF1A:TC (IC2).** The figure describes the two refined conformations, IC2A and IC2B, resulting from 3D classification. These two conformations essentially differ by the position of aIF2 with respect to h44 (see text). **a** Cryo-EM map of IC2A at a nominal resolution of 4.2 Å. **b** Cryo-EM map of IC2B at a nominal resolution of 3.3 Å. Regions of the maps are colored by component to show the 30S subunit (pale yellow), aIF1A (orange), Met-tRNA$_i^{Met}$ (yellow), aIF2γ (green), aIF2β (gray), and aIF2α (cyan). The same color code is used in all figures. The mRNA is not visible in this orientation. The view was drawn using Chimera[90]. For panel **b**, a composite made of the IC2B map contoured at two different levels is represented in order to allow visualization of the IFs together with a sufficient level of details for the 30S.

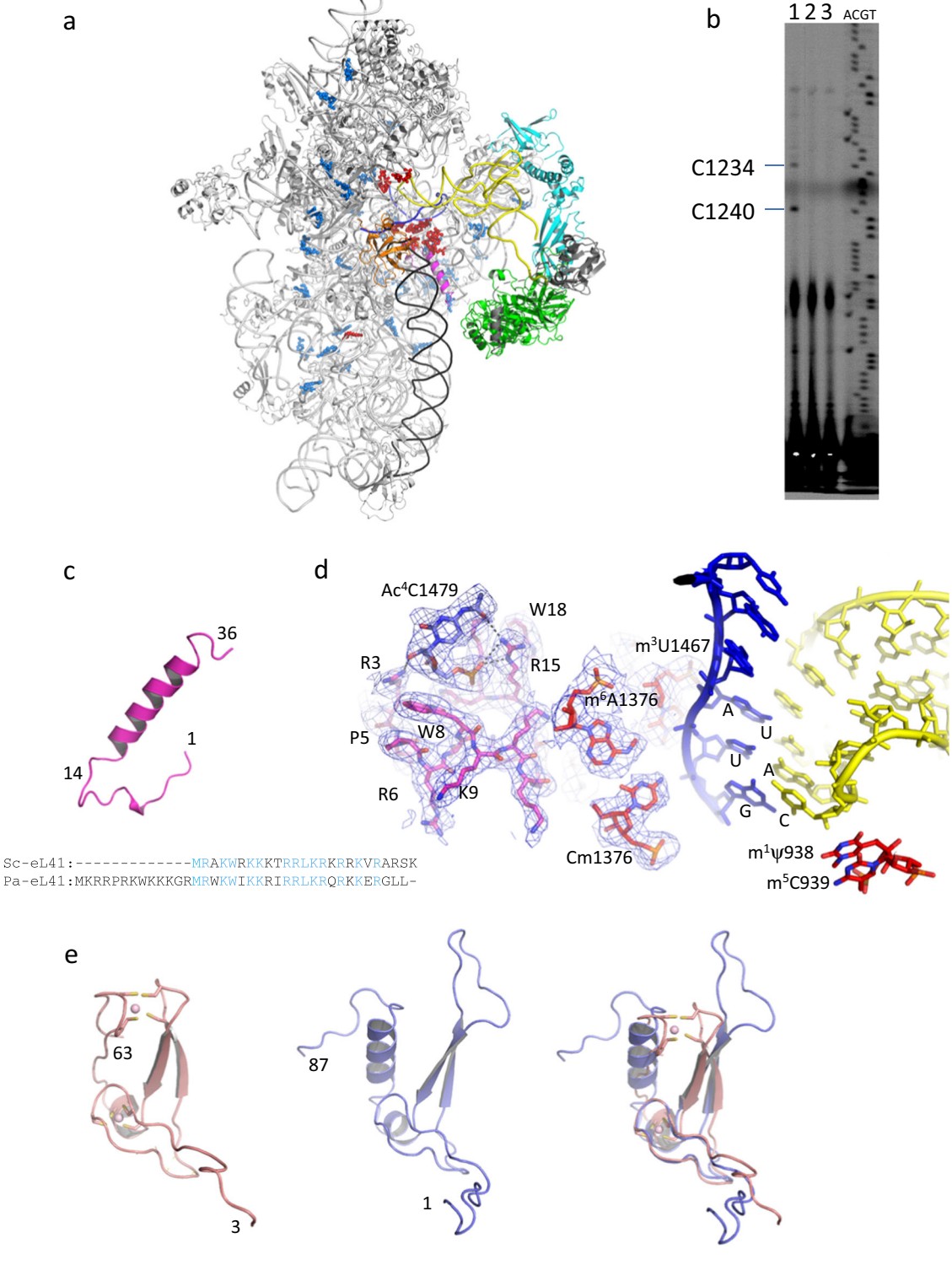

**Fig. 2 Identified archaeal specificities of Pa-30S. a** Overall view of the IC2B structure. N$^4$-acetylcytidines identified in this study are shown in blue spheres and other nucleotide modifications are in red spheres. eL41 is in magenta and h44 is in black. The view shows that N$^4$-acetylcytidines are distributed throughout the 16S rRNA. **b** Representative primer extension analysis of 16S rRNA to map ac$^4$C residues in 16S rRNA as described[32]. Lane 1, 16S rRNA treated with NaBH$_4$ (100 mM, 37 °C, 1 h), lane 2 control 16S rRNA (37 °C, 1 h), lane 3, intact 16S rRNA. The RT stops are indicated to the left of the view. RT primer is number 8 (Supplementary Table 4). Other gels are shown in Fig. S10. **c** Structure of Pa-eL41. The newly identified N-terminal extension (1–14) and the eukaryotic-archaeal core helix are shown in magenta. A sequence alignment of *S. cerevisiae* and *P. abyssi* eL41 is shown below. **d** The N-terminal extension of eL41 is shown in magenta sticks. Modified bases are shown in red sticks except ac$^4$C1479 shown in blue sticks. mRNA is in blue and tRNA is in yellow. Start codon and anticodon nucleotides are labeled. **e** Structure of the archaeal S21 ribosomal protein (pink, left) compared to eS21 (blue, middle, PDB: 6FYX). The two superimposed structures are shown on the right view. The resulting sequence alignment is shown below with structurally superimposed regions underlined. Figures 2–4 were drawn with Pymol[98].

are shown in Table 1 and in Supplementary Fig. 5, respectively. Overall, the structure is similar to that of *Pyrococcus furiosus* 30S[20]. However, many additions have been rendered possible by the higher resolution of the electron density maps. Two peculiarities concerning the ribosomal proteins deserve special attention. The first one concerns archaeal eL41 (the 2014 system for naming ribosomal proteins is used throughout[9]). As previously noted from the yeast 80S structure[21], eL41 is more strongly associated to the small ribosomal subunit than to the large one. Consistent with this observation, eL41 was indeed found in the structures of several 40S[17,22]. In the present archaeal structure, eL41 is also found in the small ribosomal subunit, as in[12]. However, in contrast with previous annotations of the genomes of *P. furiosus* or *P. abyssi*, the electron density shows that eL41 has an N-terminal extension (Fig. 2c). The protein contains 37 amino acids with an N-terminal peptide, buried in a cavity lined by h27, h44, and h45, followed by a 20 residue long helix (Fig. 2c, d and Supplementary Fig. 6a). A protein of this length has indeed been annotated as eL41 in several archaeal genomes, arguing in favor of a conservation of the N-terminal extension in this domain of life (Supplementary Fig. 6b, c). In eukaryotes, eL41 is most frequently annotated as a 25 aminoacid protein only containing the α-helix part. The functional implications related to eL41 variability will be discussed further below. Secondly, a protein found at the position corresponding to eS21 in eukaryotic ribosomes was built and identified among translation-related proteins in the annotated *P. abyssi* genome (accession number WP_010867153.1, Fig. 2e and Supplementary Fig. 7a). The structural topology of this protein resembles that of eS21. The two proteins superimpose with an RSMD of 2.1 Å for 47 aligned residues. However, sequence identities in the structurally superimposed regions are limited to five residues (Fig. 2e). In particular, the archaeal protein contains two zinc knuckles that are not observed in the eukaryotic version. Overall, a common ancestor for the archaeal and eukaryotic versions of S21 is rather difficult to envisage. In this sense, aS21 might be an archaeal-specific ribosomal protein. The widespread conservation of the protein in archaeal genomes argues in favor of this idea (Supplementary Data 1).

In 16S rRNA, we identified 44 rRNA modifications (Tables 2 and 3, Fig. 2a and Supplementary Figs. 8 and 9). Some of these rRNA modifications have already been classified as universally conserved and are clustered in the P site[17,23–26] (Fig. 3a, b). They correspond to $m^3U1467$ and the two dimethyladenosines $m_2^6A1487$, $m_2^6A1488$ (Table 4; *P. abyssi* numbering is used unless otherwise stated). Dimethyladenosines at the corresponding positions have been identified in *Haloferax volcanii* 16S rRNA[27] and the KsgA/Dim1 family of enzymes responsible for the modifications is conserved throughout evolution[28] with very few exceptions[29]. Some bacterial specific modifications are not observed but, in contrast, some modifications observed in human and *S. cerevisiae* are seen (Table 4). Overall, the pattern of P site modifications in the euryarchaeal ribosome described here appears closer to that of the eukaryotic ribosome than to the bacterial one. Importantly, we confirmed the presence of all rRNA modifications introduced in the model (Tables 2 and 3) using liquid chromatography/high-resolution mass spectrometry (LC-HRMS) (Supplementary Fig. 10, Supplementary Table 2 and Methods). The only exception is $hm^5C$, tentatively modeled at position 1378 (Supplementary Fig. 8 and Table 3).

Beside rRNA modifications already characterized in other domains of life, the electron density showed the presence of many N4-acetylcytidines (ac4C; Table 2, Supplementary Figs. 5 and 9) distributed throughout the 16S rRNA. A total of 34 N4-acetylcytidines were identified. The high level of posttranscriptional

**Table 2 N4-acetylcytidines in *P. abyssi* 16S rRNA of IC2.**

| rRNA structural element | Position |
|---|---|
| h1-uS5 | ac4C17* |
| h4 | ac4C53* |
| h11 | ac4Cm250* |
| h11 | ac4C286* |
| h12 | ac4C303* |
| h12 | ac4C319* |
| h15 | ac4C379* |
| h15 | ac4C394* |
| h18 | ac4C479+ |
| h18 | ac4C511# |
| h20 | ac4C546# |
| h21 | ac4C590* |
| h22 | ac4C626* |
| h22 | ac4C636* |
| h23 | ac4C648* |
| h22 | ac4C703+ |
| h22 | ac4C718* |
| h20 | ac4C731# |
| h24 | ac4C751* |
| h26 | ac4C828* |
| h26 | ac4C839* |
| h25-eS12 | ac4C848* |
| h19-eS12 | ac4C851* |
| h27 | ac4C868* |
| h32 | ac4C957* |
| h33 | ac4C998§ |
| h34 | ac4C1028* |
| h36 | ac4C1041* |
| h40 | ac4C1147* |
| h34 | ac4C1184* |
| h32 | ac4C1193* |
| h41 | ac4C1233* |
| h34 | ac4C1239* |
| h45-eL41 | ac4C1479+ |

34 modifications have been modeled, 19 in the body and 15 in the 30S head.
*Position of N4-acetylcytidine confirmed by RT experiments.
§Position of N4-acetylcytidine deduced from RT experiments but for which structural information is missing (end of the beak).
+position not analyzed by RT. Note that an ac4C is found at the position corresponding to C1479 in human and yeast 18S rRNA (see text).
#Position for which no arrest signal was observed in RT experiments. This may be due to lower reactivity toward sodium borohydride reduction.
Note that the presence of ac4C was also shown by LC-HRMS (Supplementary Table 2).

**Table 3 16S rRNA modifications other than N4-acetylcytidines in *P. abyssi* 30S.**

| rRNA structural element | Position |
|---|---|
| Hairpin loop:h13 | Am373 |
| P site | m1Ψ938 |
| P site | m5C939 |
| P site-uS13 | m5C1202 |
| P site-mRNA | Cm1376 |
| P site | hm5C1378 |
| P site-rsmE | m3U1467 |
| P site bulge loop:h44 | m6A1469 |
| P site-KsgA universal | m6₂A1487 |
| P site-KsgA universal | m6₂A1488 |

All modified nucleotides were observed by LC-HRMS, except for hm5C, which was tentatively modeled at position 1378. Note that an unknown modification was identified at the corresponding position in some archaea[48].

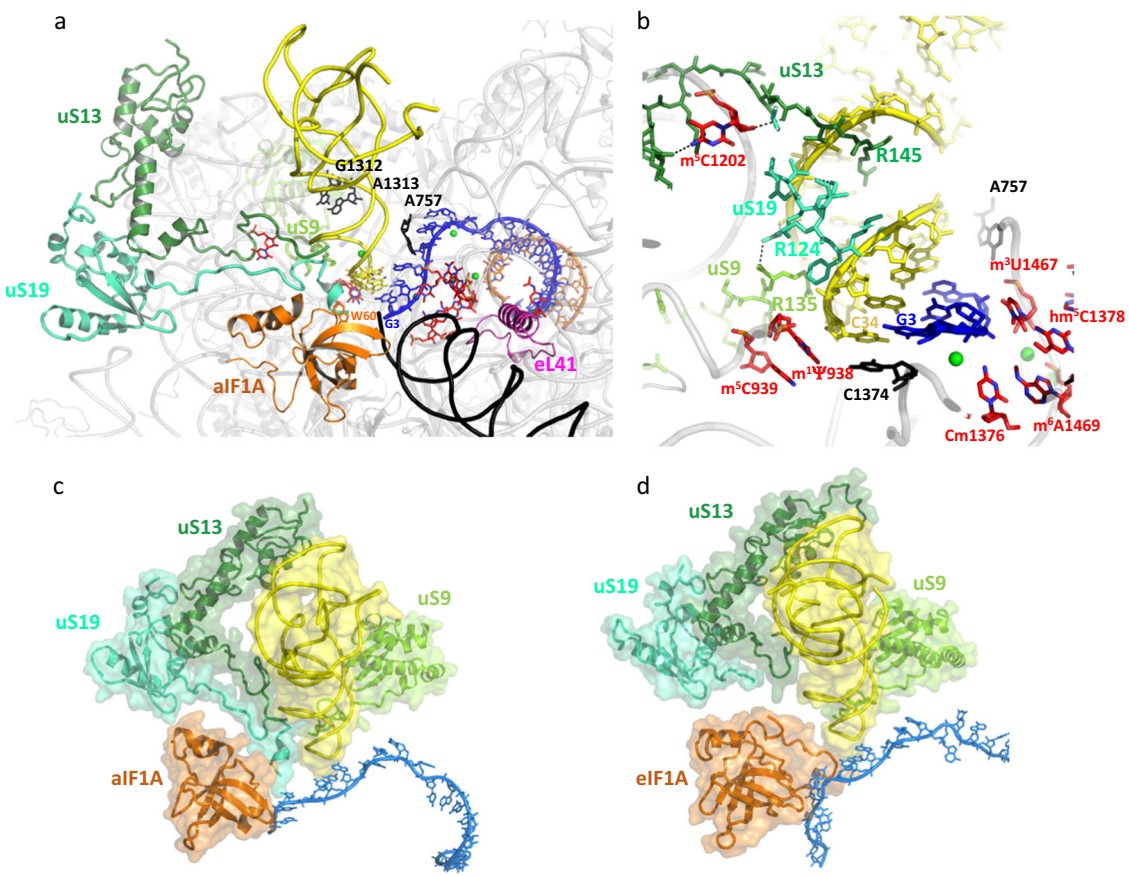

**Fig. 3 Initiator tRNA interactions at the P site. a** Overall view of protein and RNA interactions with P site tRNA in IC2B. The color code is the same as in Figs. 1 and 2. Modified rRNA nucleotides are in red and unmodified nucleotides are in black. uS13 is shown in dark green, uS19 in cyan green and uS9 in light green. h44 is in black. Putative magnesium ions are in green spheres. For clarity, aIF2 subunits have not been drawn. **b** Closeup of P site interactions. **c** View of IC2B P site tRNA and surrounding universal proteins uS9, uS13, and uS19. **d** View of P site tRNA from a eukaryotic PIC obtained in the absence of eIF1 shown in the same orientation as view c (PDB: 6FYX[17], eIF5 was not drawn for clarity).

**Table 4 16S rRNA modifications in P site vicinity in the three domains of life.**

| E. coli | T. thermophilus | P. abyssi | S. cerevisiae | H. sapiens |
|---|---|---|---|---|
| m$^2$G966 (B) | m$^2$G944 (B) | m$^1\Psi$938 (AE) | m$^1$acp$^3\Psi$1191 (AE) | m$^1$acp$^3$ $\Psi$1248 (AE) |
| m$^5$C967 | C945 | m$^5$C939 | C1192 | C1249 |
| C1228 | C1210 | m$^5$C1202 | C1461 | A1522 |
| C1400 | m$^5$C1383 | C1374 | C1637 | C1701 |
| m$^4$Cm1402 (B) | m$^4$Cm1385 (B) | Cm1376 (AE) | Cm1639 (AE) | Cm1703 (AE) |
| C1404 | m$^5$C1387 | hm$^5$C1378 | C1641 | C1705 |
| m$^5$C1407 (B) | m$^5$C1390 (B) | C1381 | C1644 | C1708 |
| **m$^3$U1498** | **m$^3$U1476** | **m$^3$U1467** | U1761 | **m$^3$U1830** |
| A1500 | A1478 | m$^6$A1469 (AE) | A1763 (AE) | m$^6$A1832 (AE) |
| C1510 | C1488 | ac$^4$C1479 (AE) | ac$^4$C1773 (AE) | ac$^4$C1842 (AE) |
| m$^2$G1516 (B) | G1494 (B) | G1485 | U1779 | U1848 |
| **m$^6_2$A1518** | **m$^6_2$A1492** | **m$^6_2$A1487** | **m$^6_2$A1781** | **m$^6_2$A1850** |
| **m$^6_2$A1519** | **m$^6_2$A1493** | **m$^6_2$A1488** | **m$^6_2$A1782** | **m$^6_2$A1851** |

Data presented in this table are from ref. [25] for E. coli[26], for T. thermophilus, and[23, 24, 99] for S. cerevisiae and human. Modified bases found in all domains of life, shown with a blue background in Supplementary Fig. 5, are in bold. Modified bases found in bacteria are indicated with (B). Modified bases found in eukaryotes and archaea, shown with a green background in Supplementary Fig. 5, are indicated with (AE).

modifications and the presence of N$^4$-acetylcytidines observed here is reminiscent of previous studies concerning two hyperthermophilic crenarchaea (*Sulfolobus solfataricus* and *Pyrodictium occultum*)[30,31]. In order to confirm the positions of ac$^4$C in the 16S rRNA sequence, we performed reverse transcription mapping on borohydride-reduced 16S rRNA as described[32] (Fig. 2b and Supplementary Fig. 11). N$^4$-acetylcytidine modifications

systematically target the second cytosine of a 5′ CCG 3′ sequence inside or at extremities of 16S rRNA helices (Supplementary Fig. 5). Remarkably, this is also true for previously identified N$^4$-acetylcytidines in eukaryotic 18S rRNA[24,33,34]. However the CCG motif was not highlighted at this time because too few modified sites were available. Ac$^4$C have been modeled in their preferred proximal conformation as observed in the ac$^4$C nucleoside crystal

structure[35] and also as calculated using quantum chemistry methods[36]. This conformation allows canonical Watson-Crick base pairing interactions. Moreover, the acetyl group reinforces π–π stacking with adjacent bases that increases stability of the base pair. Therefore, this modification distributed throughout the 16S rRNA might largely contribute to the hyperthermostability. Some ac[4]C residues also interact with ribosomal proteins. One notable interaction involves the eukaryotic-conserved ac[4]C1479 and R15 of eL41 (Fig. 2d). This position corresponds to ac[4]C1773 and ac[4]C1842 previously characterized in *S. cerevisiae* and in human ribosomes, respectively[24,33,34]. An archaeal orthologue of the eukaryotic Nat10/Kre33 enzyme could be responsible for these modifications. Taking into account the small RNA guided mechanism of Nat10/Kre33 modifications of 18S rRNA, we hypothesized that small RNA guides would assist Nat10/Kre33 in archaea. Small RNAs would also explain the non-systematic modification of CCG sequences. Finally, because conservation of the corresponding CCG sequences in archaeal 16S rRNA sequences is not obvious (http://www.rna.icmb.utexas.edu/), the presence of N[4]-acetylations in other archaea has to be studied in each case.

**Structure of the mRNA.** An mRNA corresponding to the natural start region of the highly expressed elongation factor aEF1A from *P. abyssi*[12,14] was used in IC2. It contains a strong SD sequence with a spacing of 10 nucleotides to the AUG start codon (Fig. 4a and ref. [37]). The SD duplex is extended to 9 nucleotides and involves the 5′AUCACCUCC 3′ sequence of the 3′-end of the 16S rRNA (Fig. 4 and Supplementary Fig. 7b). The SD helix is positioned in the exit chamber delineated by uS11, eS3, and h26 on the one side and by uS7, eS28, h28, and h37, on the other side (Fig. 4b). Interactions of uS11 with eS28 and uS7 connect the platform to the head and form the SD duplex channel. uS2 is located at the end of the chamber. Downstream from the SD duplex, the mRNA goes towards the E site. A single unpaired base provides the junction (Fig. 4a). It is stabilized by the tip of the β-hairpin of uS7, as already observed in bacteria and eukaryotes[17,38,39]. This part of uS7 can serve as a gate to the E site. Downstream from U$_{-4}$, the mRNA makes a sharp turn and the bases of the E, P, and A codons are pre-positioned in triplets with a kink between the adjacent codons (Fig. 4 and Supplementary Fig. 7b). During translation elongation, the mRNA kink would be important for reading frame maintenance and to prevent mRNA slippage[40,41]. Finally, the structure of the mRNA remains identical in the two IC2 conformations.

**The initiator tRNA is stably bound to the P site.** In IC2A and B, the initiator tRNA is stably bound to the P site and base paired with the mRNA AUG start codon. The interactions of the initiator tRNA within the P site will hereafter be described from the IC2B structure because of its higher resolution.

A set of rRNA modifications contribute to the stabilization of the codon:anticodon duplex (Fig. 3a, b) as previously observed in bacteria[25,26,42,43] and eukaryotes[17,23,24]. The base pair C34$_{tRNA}$:G3$_{mRNA}$ is stabilized by two types of stacking interactions. On the one hand, C1374 (C1400 in *E. coli*) is stacked onto the base pair and, on the other hand, m[1]ψ938 (m[2]G966 in *E. coli*) is stacked onto the ribose of C34. Interestingly, a U is systematically found at the latter position in the sequences of small ribosomal subunit (SSU) rRNA of archaea and eukaryotes (http://www.rna.icmb.utexas.edu/[44]), whereas a G is encountered in bacteria. In eukaryotes, this uridine is hypermodified to m[1]acp[3]ψ[17,24,45,46]. Here, we modeled an m[1]ψ taking into account its presence at this position in *M. jannaschii*[47] and the occurrence in the *P. abyssi* genome of an orthologue of the corresponding modification enzyme Nep1[46] (Supplementary Fig. 8). Even if the presence of

the 3-amino-3-carboxypropyl in position 3 of the m[1]ψ938 has been shown in *H. volcanii*[27,48], this chemical group is not visible in the density and no ortholog of the modification enzyme Tsr3 could be found in *P. abyssi*. This reflects some variability of the set of rRNA modifications within the archaeal phyla[31]. Notably, the correct orientation of m[1]ψ938 is due to stacking onto m[5]C939 (Fig. 3b). Methylation of C939 is further supported by the identification in the *P. abyssi* genome of an ortholog of the RsmB enzyme responsible for the corresponding modification in bacteria (Accession number WP_010868369, NCBI database, see also[49]). On the side of the codon, the ribose groups of A1 and U2 are stacked against m[3]U1467 (m[3]U1498 in *E. coli*). The G1 phosphate group is held in place through interaction with the m[6]A1469-Cm1376 pair. The codon–anticodon helix is also stabilized by two magnesium ions, one bridging the phosphate group of A37 with Cm32 and A38 and the other bridging the phosphate groups of A and U bases of the start codon. Magnesium ions were also observed at similar positions in the bacterial 70S[50]. A second layer of rRNA modifications made up of m[6]$_2$A1487, m[6]$_2$A1488, hm[5]C1378, stabilizes the first layer. Notably, the N-terminal part of eL41, embedded between h27, h44, and h45 rRNA helices, contacts several modified bases linked to the P site (m[6]$_2$A1488, hm[5]C1378, m[6]A1469). Moreover, ac[4]C1479 interacts with R15 from eL41 (Figs. 2d and 3a). This interaction, conserved in eukaryotes, also contributes to stabilize eL41 in the cavity. Finally, the interaction of the C-terminal part of eL41 with aIF2γDIII may provide a physical link between the P site and the aIF2γ:h44 contact region (Supplementary Fig. 4). In the anticodon stem, type II and type I A-minor interactions involving the GC base pairs 29–41 and 30–40 with G1312, A1313 (G1338, A1339 in *E. coli*) are observed. On the other side, the pocket is delineated by A757 (A790 in *E. coli*) from h24 loop. These interactions (Fig. 3a, b) are conserved in bacteria[40,51] and eukaryotes[17] and are therefore universal.

The C-terminal tails of the three universal proteins, uS9, uS13, and uS19 contact the initiator tRNA. The universally conserved C-terminal arginine of uS9 is hydrogen bonded to the phosphate groups of Cm32, U33, A35, like in eukaryotes[17] and bacteria[25,26,40,51]. Here, the position of the C-terminal arginine uS9-R135 is further stabilized by hydrogen bonds between the carboxylate group and uS19-R124 (Supplementary Fig. 7c). We modeled the C-terminal tail of uS13 entering the major groove of the tRNA anticodon stem, with the guanidinium group of uS13-R145 facing the Hoogsteen edge of G30. The electron density for the three terminal lysine residues of uS13 is missing. Interestingly, the guanidinium group of uS13-R145 occupies a position corresponding to that of A37 in its "unstacked" conformation observed in the anticodon loop of free initiator tRNA[13,52]. One may imagine that during tRNA accommodation, R145-uS13 selects G30-C40 in the 3 G-C pairs major groove and facilitates the motion of A37 towards the anticodon loop where it is stacked against U36 and A38. On the same side of the tRNA molecule, just below uS13, is the uS19 C-terminal extremity. Side chains of uS19-T123 and uS19-S125 interact with the phosphate group of G30 and uS19-R124 is stacked against the phosphate backbone from tRNA-G29 to tRNA-G30 (Fig. 3b and Supplementary Fig. 7c). In uS19, a short α-helix (residues 125–129) follows R124 (Fig. 3a). Finally, faint density for the three uS19 C-terminal residues (130–132) suggests their positioning at the edge between the P and A site codons, close to W60 of aIF1A (Fig. 3a and Supplementary Fig. 7c).

## Discussion
In a previous study, we determined the cryo-EM structure of an archaeal 30S initiation complex containing all factors involved in

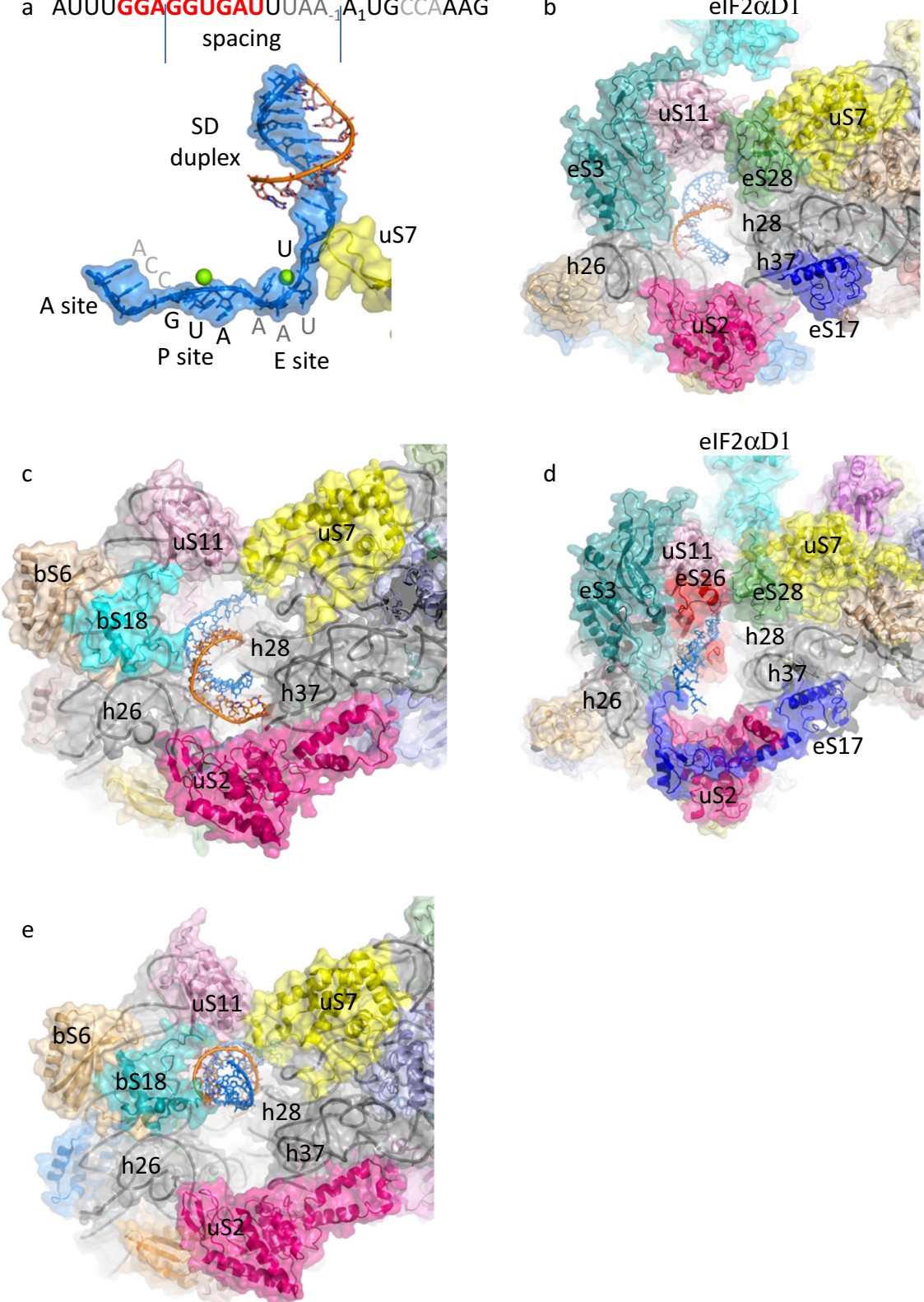

**Fig. 4 Structure of the mRNA and comparison of the mRNA exit channels in the three domains of life. a** Surface representation of IC2B mRNA in blue. The anti-SD sequence is shown in orange sticks. The view emphasizes the sharp turn of the mRNA downstream from $U_{-4}$ and the kink between adjacent codons at the A, P, and E sites. The β-hairpin of uS7 is in yellow. Magnesium ions are shown in green spheres. **b** Surface representation of the mRNA exit channel in IC2B. Ribosomal proteins and rRNA helices are labeled. mRNA is in blue and the anti-SD sequence is in orange. The color code is the same for the four **b–e** views. The four structures are shown in the same orientation. **c** Surface representation of *T. thermophilus* 30S bound to an mRNA allowing a "free choice" of the start codon (PDB: 4V4Y[83], Supplementary Table 3). **d** Surface representation of *K. lactis* 40S bound to a 49-nt unstructured mRNA (PDB: 6FYX[17]). **e** Surface representation of *T. thermophilus* 30S bound to an mRNA containing the **GGAGGU**AAAA**AUG** sequence (PDB: 5LMQ[51]).

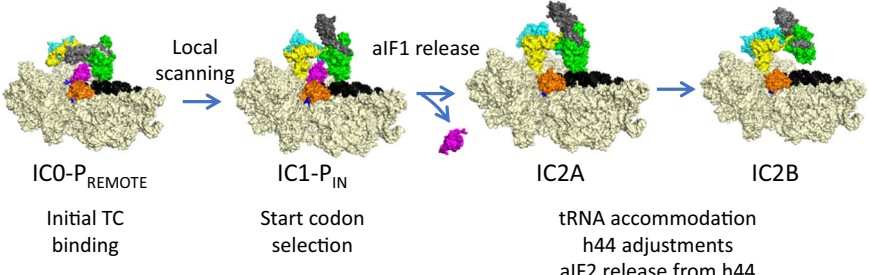

**Fig. 5 Steps of translation initiation in *P. abyssi*.** Surface representation of successive *P. abyssi* translation initiation complexes. IC0-P$_{REMOTE}$ and IC1-P$_{IN}$ are from ref. [12]. tRNA accommodation, h44 adjustments and aIF2 release from h44 are associated with the release of aIF1. After start codon recognition, aIF1 is released and the initiator tRNA is fully accommodated. Release of aIF1 would cause both Pi release from aIF2γ and h44 adjustments leading to irreversible aIF2 release (see discussion).

start codon selection (aIF1, aIF1A, and aIF2). In the major conformation, IC0-P$_{REMOTE}$, the anticodon stem–loop of the tRNA is out of the P site (Fig. 5). The ternary complex structure is similar to that of the free one, showing that it is not constrained by the ribosome. aIF2γ is bound to h44 and interacts with aIF1. In the second conformation, called IC1-P$_{IN}$, the anticodon stem–loop of the initiator tRNA is bound to the P site, while the position of aIF2γ on h44 has not changed. These two positions are in equilibrium and the transition from one position to the other, accompanied by a 30S head motion, reflects the dynamics of the PIC during testing for the presence of a start codon in the P site[12,14]. As observed for eukaryotic PIC, stabilization of the tRNA in the P site is impaired by aIF1[53,54]. Consistent with this idea, IC2, made in the absence of aIF1, shows a more homogeneous conformation of the tRNA in agreement with its enhanced stability suggested by toeprinting experiments[14].

The IC2A and IC2B conformations were compared to IC0-P$_{REMOTE}$ and IC1-P$_{IN}$[12] by superimposing the 30S bodies of the four structures. In IC2, aIF1A is bound to the A site, though its position is less buried than that observed in IC0 and IC1. In IC2, the conformation of h44 in the vicinity of the binding sites of aIF1 and aIF1A has changed (Supplementary Fig. 12). In IC2A, the conformation of the TC resembles that seen in IC1 (Fig. 5). However, in IC2A, the initiator tRNA is slightly displaced toward the "70S P site" position[26] and aIF2γ is slightly repositioned on the side of the vacant aIF1 binding site. aIF2γDIII still contacts the G1391-A1392 h44 bulge. In IC2B, TC is detached from h44 and its conformation is relaxed, close to the conformation observed in the crystal structure[55] and in IC0-P$_{REMOTE}$.

In eukaryotes, it was shown that full release of eIF2 is linked to the release of Pi coming from GTP hydrolysis on eIF2[56]. In archaea, we previously described possible contacts between the N-terminal domain of aIF1 and the switch regions controlling the nucleotide state of aIF2γ[12]. These contacts may link aIF1 release to Pi release. Because we used GDPNP in IC2 complex preparation, aIF2 is not fully released and contacts between aIF2 and the acceptor stem of the initiator tRNA still exist. Motion of h44 close to the P site may be caused by aIF1 release (Supplementary Fig. 12), as previously proposed in eukaryotes[18,19]. Readjustments of the position of h44 in the bulge region could explain the relaxation of the contacts between this helix and γDIII. Both these movements and the release of contacts between aIF1 and aIF2γ observed in IC0 and IC1 would explain how aIF2 is detached from the ribosome after start codon recognition and aIF1 release. Interestingly, contacts between h44 and other translation factors were observed, such as those involving ABCE1 during recycling[57]. Therefore, conformational adjustments of h44 may be a general mechanism controlling binding and release of factors during the translation cycle.

In IC2A and B, the anticodon stem of the tRNA is tightly bound to the P site partly thanks to interactions with the C-terminal tails of the universal proteins uS9, uS13 and uS19. The role of uS9 C-tail in fidelity was previously shown by studies with bacterial[58–61] and yeast systems[62,63]. Moreover, in eukaryotes, uS9 was shown to favor the recruitment of the TC on the ribosome[62]. The present archaeal structure suggests that uS9 tail is universally related to fidelity. The C-tail of uS13 was only modeled in a bacterial IC[51] and the C-tail of uS19 was, to our knowledge, never observed in initiation complexes before this study. In the present structure, uS13 and uS19 interact with G30 of the second base pair of the almost universally conserved three GC base pairs in the initiator tRNA anticodon stem (Fig. 3a, b and Supplementary Fig. 7c). The present observations in the archaeal system agree with previous studies in *E. coli* showing that the central GC pair and particularly base G30 was the most crucial nucleotide for translation fidelity[64–66].

Interestingly, several studies in yeast identified allosteric information pathways connecting functional centers in the large ribosomal subunit (LSU) to the decoding center in the SSU through the B1a and B1b/c intersubunit bridges[21,67,68]. In eukaryotes and archaea, uS13 participates in B1b/c bridge and uS19 is part of the B1a bridge. One can therefore imagine that some molecular information is relayed through these two proteins to facilitate LSU joining after the accommodated state of the initiator tRNA in the P site has been sensed by the C-tails of uS13 and uS19.

Around the P site, we observed a series of rRNA modifications. In bacteria, m$^2$G966 and m$^5$C967 have been shown to participate in fine-tuning of initiation[58,69]. The variations in the three domains of life of the residue corresponding to *E. coli* 16S rRNA-G966 and of its modifications (Table 4) are likely related to evolution of translation initiation. eL41 is connected to the network of rRNA modifications close to the P site by the bulge loop between h44 and h45. Importantly, the C-terminus of eL41 also contacts h44 and aIF2γDIII. eL41 may therefore relay some structural information from the P site to the γDIII binding site and participate in the control of aIF2 release after start codon recognition. The question of the phylogenetic conservation of eL41 is rather puzzling. As evidenced here, archaeal representatives contain 37 residues (Supplementary Fig. 6c). However, eL41 has not been identified in all archaeal genomes[8]. One possibility is that its identification has been hampered by the reduced size of the protein. In higher eukaryotes, eL41 has been annotated as a 25 aminoacid protein and no supplementary N domain is observed in eukaryotic cryo-EM structures[17,24]. Notably, in some lower eukaryotes such as *Plasmodium falciparum*[70], eL41 is longer. In bacteria, no orthologue of eL41 has been identified and the corresponding site is vacant on the ribosome. However, a recent cryo-EM study identified a 33 aminoacid residues protein named

bS22 conserved in actinobacteria located at the eL41 binding site[71]. Finally, in human and yeast mitochondria, the N-terminal part of the mitochondria-specific mS38 protein occupies this position[72–74]. mS38 is proposed to be related to translation of mitochondrial mRNAs lacking typical SD sequences[74]. Overall, the whole data indicate that variability of eL41 might be related to evolution of translation initiation.

The present structure was also compared to the structure of a *Kluyveromyces lactis* initiation complex in which the N-terminal domain of eIF5 is found at the location vacated by eIF1, in front of the anticodon stem of the initiator tRNA[17]. This Kl-PIC structure represents a step of initiation occurring after start codon recognition. The initiator tRNA is stabilized by interactions with eIF5-NTD and the N-terminal extension of eIF1A. As compared to the present structure, the initiator tRNA adopts a position slightly more tilted towards the head of the SSU. Interestingly, eIF1A is more deeply bound to the A site and its N-terminal extension occupies the position observed here for the C-terminal tail of uS19. Moreover in Kl-PIC, the C-terminal tail of uS13 is not visible (Fig. 3c, d). According to what is observed in IC2, a step following the one illustrated by the Kl-PIC structure[17] may be a repositioning of eIF1A and a relocation of the C-terminal tails of uS13 and uS19 to further stabilize the initiator tRNA in its accommodated state preceding eIF5B loading.

In bacteria and in some archaea, it has been demonstrated that the SD sequence played an important role in the formation of the IC by base-pairing with the anti-SD sequence at the 3′ end of 16S rRNA[37,75–77]. In eukaryotes however, canonical translation involves scanning of the PIC searching for AUG in an optimal context[78,79]. The current structure highlights similarities and differences in the exit mRNA channel in the three domains of life (Fig. 4). Overall, the euryarchaeal exit tunnel observed here is of the eukaryotic type (Fig. 4b, d). uS11, eS3, and h26 are located on one side of the cavity whereas uS7, eS28, h37, and h28 are on the opposite side. Two notable differences are, however, observed. First, in eukaryotes, eS17 has a long C-terminal extension contacting the mRNA. Second, eS26 stabilizes the 3′ end of the mRNA[17]. Interestingly, eS26 was proposed to be involved in recognition of Kozak sequence elements[80]. Moreover, IFs are involved in stabilization of the mRNA in the exit channel in eukaryotes. Indeed, Kozak consensus nucleotides are recognized in the E site by domain 1 of eIF2α. In addition, eIF3a subunit would also stabilize the mRNA at the exit channel pore[17]. These differences illustrate how eukaryotic and euryarchaeal ribosomes evolved different binding modes of the mRNA in the exit pocket, in relation with the canonical eukaryotic scanning mode vs. the SD-assisted AUG recognition mode occurring in many genes in the archaeal domain. In this view, it is notable that eS26 is absent in euryarchaeotes but present in crenarchaeota/lokiarchaeota genomes[7,81]. Because the archaeal version of the exit chamber is a simplified version of the eukaryotic one, this argues in favor of the controversial hypothesis that eukaryotic ribosomes have evolved from within the archaeal version[10,82].

When compared to all available bacterial ribosomal complexes, the present position of the SD duplex in the chamber corresponds to the down position observed in[83] where an mRNA designed to allow a "free choice" of the start codon, with a 12 nucleotide spacing, was used (Fig. 4 b, c) or to the SD_in ("stand-by") position defined in ref. [84] where an mRNA GAAAGA lacking the upstream region was used. In contrast, when the classical model mRNA, based on the phage T4 gene 32 mRNA, chosen for its high stability, was used, the SD duplex adopted the "up" tense position in the chamber[38,41,51] (Fig. 4b, d and Supplementary Table 3). Notably, this model mRNA has a spacing of seven nucleotides instead of ten in the mRNA used in this study. Hence, the structure of the mRNA observed here would represent a

relaxed state favorable to translation initiation efficiency, as expected for an abundantly translated mRNA such as the aEF1A one. The distance between the chamber and the start codon may act as a ruler leading to translation initiation regulation according to the spacing between the SD and the start codon. Comparison of the archaeal exit channel with the bacterial one shows that bS6 and bS18 are found in place of eS3. eS28 is absent in bacteria and uS2 possesses a supplementary C-terminal domain, at the location of eS17. Therefore, archaeal and bacterial exit channels appear as two structural solutions for binding the SD duplex. The spacing between the AUG codon and the SD sequence changes the position of the duplex in the chamber, probably explaining how it influences translation initiation efficiency.

The present study of an archaeal initiation complex fills a gap in high-resolution structures of ribosomes representative of the three domains of life. It provides new structural information useful in support of evolution models based on sequence and experimental data. Universal elements located in the SSU common core at the decoding center ensure stabilization of universal features of the initiator tRNA. The mRNA exit chamber that does not make part of the SSU core[85] indeed shows evolution related to domain-specificities for mRNA binding. On the other hand, the present study also demonstrates the occurrence of a large number of N⁴-acetylcytidines in the rRNA of a hyperthermophilic organism. N$^4$-acetylation of cytidines is of increasing interest since its discovery in mRNA coding sequences where it promotes stability and translation efficiency[86,87]. Our study shows that the targeted sequence in the 16S rRNA from *P. abyssi* is systematically Cac$^4$CG. This rule also appears to apply for the ac$^4$C identified to date in 18S rRNA. In this view, our findings also bring information important to understand a possible general mechanism of ac$^4$C modification.

## Methods

**IC2 complex preparation and cryo-EM analysis**. The strategy used to prepare the IC2 complex was adapted from a previously described one[12]. In brief, archaeal 30S subunits from *P. abyssi* (Pa-30S) were purified and mixed with IFs Pa-aIF1A, the ternary complex Pa-aIF2:GDPNP:Met-tRNA$_i^{Met}$A$_1$-U$_{72}$ and a synthetic 26 nucleotide-long mRNA corresponding to the natural start region of the mRNA encoding the elongation factor aEF1A from *P. abyssi* (A(−17)UUU**GGAGGU-GA**UUUAAA(+1)**UG**CCAAAG(+9))[13]. The complex was purified by affinity chromatography (TALON, Clontech) using N-terminally tagged versions of Pa-aIF2β and Pa-aIF2α. An excess of mRNA, aIF1A and TC was added before dilution and spotting onto Quantifoil R2/2 grids with an extra 3 nm carbon layer (Quantifoil, Inc) for cryo-EM analysis. Cryo-EM images were collected on an FEI Titan Krios microscope operated at 300 kV at the eBIC center, Diamond Light Source, England (Table 1).

**Data processing**. Image processing was performed with RELION 2.1[88]. A total of 2174 images was selected for the autopicking tool. Several rounds of 2D classification and an initial 3D refinement with 330 K particles gave a 3.3 Å resolution density map. While the density for the 30S subunit showed high-resolution details, the density allocated to aIF2 and aIF1A was less resolved. Several rounds of 3D classification, 3D refinement and particle polishing gave an intermediate density at 3.2 Å resolution with 218 K particles but the bound factors remained poorly resolved. Density subtraction of different parts of the complex gave final maps of the 30S head alone, the 30S head with associated factors and 30S body alone at 3.1, 3.7, and 3.2 Å resolution, respectively (Supplementary Fig. 1). Further 3D classification on the aIF2 binding region gave 7.2, 3.4, and 5.6 Å resolution. The three classes obtained correspond to different positions of aIF2 relative to the 30S and were called IC2A, IC2B, and IC2C (Supplementary Fig. 1). Because very weak electron density for aIF1A was observed in IC2C, this class was not considered as representative of an IC2 conformation and was therefore not further refined. Finally, the particles belonging to the IC2A and IC2B classes were re-extracted from raw images and reprocessed to obtain final density maps for IC2A and IC2B complexes at 4.2 Å (34 K particles) and 3.3 Å resolution (142 K particles), respectively, according to the 0.143 FSC gold-standard criterion (Supplementary Fig. 2). Local resolution estimation was carried out using the program RESMAP[89].

**30S model building**. A full atomic model of the *P. abyssi* 30S has been built using an iterative approach including sequence alignment with RNA and proteins of known structure (PDBs: 4V6U[20], 5JB3[12], Table 1) rigid body fitting in

CHIMERA[90], real space refinement and geometry regularization in COOT[91] combined with real space refinement in PHENIX[16]. All model components (rRNA, ribosomal proteins) were first fitted in the map densities of the 30S head or body alone with simultaneous optimization of stereochemical properties and correction of intra- and intermolecular steric clashes. Secondary structure restraints for further refinement were determined directly from the model using the web interface of 3DNA-DSSR[92] and recalculated at each round of refinement. For the three RNAs present in the models, hydrogen-bonding, base-pair and stacking restraints were applied. Adequate restraint files were constructed for modified nucleotides. The high-resolution limit was set during refinement to match the nominal resolution obtained by postprocessing in RELION. The initial structure obtained was then subjected to rigid body refinement as implemented in phenix.real-space-refine in the IC2B density map and further refined. The crystallographic structure of aIF1A from *P. abyssi* (PDB: 4MNO[12]) was manually rigid-body fitted in the cryo-EM map with COOT. Moreover, because the quality of the electron density of aIF2 did not allow side-chain fitting, we used the crystallographic structure of aIF2 from *S. solfataricus* (PDBs: 4RD4[93], 3V11[55], 2QMU[94]) to perform rigid body fitting even if aIF2 from *P. abyssi* was used during complex preparation. The model of the initiator tRNA was fully reconstructed. The anticodon stem–loop region was very well defined. In contrast, according to the higher mobility of the solvent side of the aIF2:tRNA complex, the electron density of the acceptor helix was weaker. The same strategy was applied to account for the electron density of IFs in the IC2A density map. The full IC2A and IC2B models were then subjected to cycles of systematic inspection and manual corrections using COOT followed by refinement in PHENIX. Finally, the entire structure was validated using MOLPROBITY[95] as implemented in PHENIX[16] and correlation coefficients were calculated (Supplementary Table 1). Various programs from CCP-EM[96] were also used throughout the study. The final refinement statistics are provided in Table 1 and in Supplementary Table 1.

**rRNA modifications and ribosomal proteins**. The full atomic model of the *P. abyssi* 30S allowed us to identify the N-terminal extension of eL41 and an archaeal version of eS21, as described in the Results section. rRNA modifications visible in the cryo-EM maps have been modeled taking into account information coming from high-resolution cryo-EM and X-ray structures[24–26], RNA modifications databases (http://mods.rna.albany.edu/, http://modomics.genesilico.pl/), and literature. The presence of the rRNA modifications was then verified using mass spectrometry as well as primer extension analysis for $N^4$-acetylcytidines. A total of 44 rRNA modifications have been identified including 34 $N^4$-acetylcytidines (Tables 2 and 3).

**Primer extension analysis of $N^4$-acetylcytidines**. 16S rRNA was prepared from purified *P. abyssi* 30S subunits[12] using a standard phenol–ether extraction protocol. 16S rRNA reduction was performed as described[32]. Totally, 3 μg of rRNA was incubated with either sodium borohydride (100 mM in $H_2O$) or water (control sample) in a final reaction volume of 50 μL. Samples were incubated 60 min at 37 °C, quenched with 7.5 μL of 1 M HCl and neutralized with 7.5 μL of 1 M Tris-HCl pH 8.0. Reactions were then ethanol precipitated and rRNA was resuspended in water at a final concentration of 140 ng/μL. Reverse transcription reactions were then performed as follows: 0.28 picomoles of either reduced, nonreduced (control sample) or intact 16S rRNA (140 ng) were mixed to 4 picomoles of 5′ fluorescently labeled primer (see Supplementary Table 4), 1 unit RNAse OUT inhibitor (Invitrogen) and 1 μL of 10× annealing buffer (600 mM $NH_4Cl$, 100 mM Hepes pH 7.5, 70 mM 2-mercaptoethanol) in a final volume of 10 μL. Reactions were then heated 5 min at 65 °C and cooled directly on ice. After annealing, 1.3 μL of 100 mM Mg Acetate and 0.75 μL of dNTP (3.75 mM each) were added and reverse transcription was started by adding 1 μL of AMV reverse transcriptase (3 U/μL, Promega). After 15 min incubation at 51 °C, the reaction was stopped with 3 μL of a solution containing 95% formamide and blue dextran. Reverse transcripts were analyzed on a Licor 4200 DNA sequencer. Sequencing reactions made from a PCR amplified sample of the 16S rRNA and the RT primer were loaded in order to identify the stop positions. Typical experiments are shown in Fig. 2b and in Supplementary Fig. 11.

**Mass spectrometry analysis of hydrolyzed 16S rRNA**. Fully hydrolyzed 16S rRNA suitable for mass spectrometry analysis was prepared as described[97]. Samples were then diluted in 0.1% formic acid (FA) prior to analysis. Chromatographic grade solvents (99.99% purity), acetonitrile (MeOH) and FA were purchased from Sigma Aldrich. LC-HRMS analyses were performed with the timsTOF mass spectrometer coupled with an Elute HPLC system (Bruker Daltonics, Bremen, Germany). The sample (10 μL, 2.5 μg digested 16S rRNA) was injected and separated on an Atlantis T3 column (3 μm, 150 × 2.1 mm; Waters, Saint Quentin, France). The effluent was introduced at a flow rate of 0.2 mL min$^{-1}$ into the interface with a gradient increasing from 10% of solvent B to 50% in 6 min to achieve 70% at 8 min (A: water with 0.1% FA; B: methanol with 0.1% FA). From 8 min to 12 min, the percentage of solvent increased up to 90% of B. The flow was then set at 10% of B for the last 6 min. Electrospray ionization was operated in the positive ion mode. Capillary and end plate voltages were set at −4.5 and −0.5 kV, respectively. Nitrogen was used as the nebulizer and drying gas at 2 bar and 8 L/

min, respectively, with a drying temperature of 220 °C. In MS/MS experiments, the precursor ion was selected with an isolation window of 1 Da and the collision-induced dissociation was performed using collision energies (Ecol) ranging from 7 to 25 eV. Tuning mix (Agilent, France) was used for calibration. The elemental compositions of all ions were determined with the instrument software Data analysis, the precision of mass measurement was better than 3 ppm. All nucleosides have been characterized by both molecular (MH$^+$) and fragment ions (BH$_2$+) (Supplementary Table 2). The only exception was ac4Cm for which only the MH+ ion was identified.

**Statistics and reproducibility**. The cryo-EM data reported here come from a single experiment and a single grid was used for data collection. Individual images with either bad ice, as shown by visual inspection, or too much motion or astigmatism, as shown by power spectra, were excluded from the dataset. Data collection, processing and refinement statistics (see Table 1) were calculated in RELION[88], RESMAP[89], and PHENIX[16].

**Reporting summary**. Further information on research design is available in the Nature Research Reporting Summary linked to this article.

## Data availability
The EMDataBank accession numbers for the EM maps reported in this paper are EMD-10320 (IC2A), EMD-10322 (IC2B), EMD-10323 (IC2 body), and EMD-10324 (IC2 head). The coordinates of the models fitted in the maps have been deposited in the Protein Data Bank (PDB: 6SW9, IC2A, PDB: 6SWC, IC2B, PDB: 6SWD, IC2 body, PDB: 6SWE, IC2 head).

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

## Acknowledgements

This work was supported by grants from the Centre National de la Recherche Scientifique and Ecole polytechnique to Unité Mixte de Recherche n°7654 and by a grant from the Agence Nationale de la Recherche (ANR-17-CE11–0037). We thank Alistair Siebert for data collection at the electron biological cryo-imaging facility (eBIC, Diamond Light Source, UK). We thank Thomas Gaillard for helpful discussions.

## Author contributions

P.D.C., Y.M. and E.S. designed the research; P.D.C., C.L.S., S.B., Y.M. and E.S. contributed to the experimental work; P.D.C., Y.M. and E.S. analyzed the data and wrote the paper. P.D.C., C.L.S., S.B., Y.M. and E.S. reviewed and edited the paper.

## Competing interests

The authors declare no competing interests.
