## [Peer Review File · Communications Biology]

Reviewers' comments:

Reviewer #1 (Remarks to the Author):

Manuscript by Pierre-Damien Coureux *et alia* describes electron density maps inferred by using cryogenic transmission electron microscopy imaging of purified archaeal translation initiation complexes. The electron density maps are refined to high resolutions and classified to clearly segregate into several classes. Two major classes and the corresponding initiation complex conformations (or complex types) were detectable, one with fully and another with partially accommodated methionylated initiator tRNA. The high resolution obtained in the study allowed attribution of novel feature types within the structures, such as species-specific modifications of the ribosomal RNA.

It is an excellent study. The work significantly adds to the current knowledge. While the existing PDB entries 5JB3 and 5JBH (from the same group; released in 2016) of similar archaeal translation initiation complexes outlined the overall layout of these structures, the newly refined data provides much higher resolution (~ 3.3 vs. ~ 5.3 Å). The refinement parameters are also stricter in the current manuscript. This allows for the visualisation of smaller features and adds much certainty to the structures and modelling. The manuscript is clearly written; delivery of the results is on a very high level; the flow of the text is concise and logically well-ordered. I have no doubts the work will be cited and will become an excellent resource.

I have no major criticisms or objections of any kind. I have a few technical comments or suggestions; it would be nice to see these implemented before acceptance. I can highly recommend this work for publication in *Communications Biology* generally in the current form.

Major points

1. Did the authors attempt classification *ab initio* and what was the actual reason for dropping it for the initial refinement? I also think that not enough attention was diverted to the classification. What the other types of complexes could represent? I understand that this may be the focus of subsequent studies, or the authors deemed the other types as 'uninteresting'. However, I think that the other classes of the complexes identified should be described with more detail.

Minor points

Abstract

1. While hyphenation for the description of the components of molecular complexes is acceptable (and it is a widespread practice), in my opinion, the use of colon or dot signs for the non-covalent interactions is better, as it discriminates covalent bonds or interactions from the non-covalent counterparts. For example, in the variant 'aIF2-GTP-Met-tRNA_i^{Met}' used by the authors already in the abstract, it seems that the methionine attachment to tRNA is of the same nature as the interactions with the other components. For a good structural study, replacing this with aIF2:GTP:Met-tRNA_i^{Met} or aIF2•GTP•Met-tRNA_i^{Met} (depending on the style preferences) could be a desirable improvement.

Introduction

2. In the introduction (and other parts), I suggest replacing methionyl-initiator tRNA with methionylated initiator tRNA.

3. I suggest explicitly specifying 'assist the selection mechanism' -> 'assist the start codon selection mechanism'.

4. Inconsistent five or three prime sign use (sometimes apostrophe, sometimes prime sign).

5. I am unsure if the 'spring force' is the best wording; is it purely an elastic deformation (changes in the bonds lengths/angles), or there is an entropic component as well in this case (mobility restriction)?

Results

6. I would probably suggest using 'n-n interactions' instead of 'pi interactions'.

Figure Legends

7. In Fig. 1 legend, I suggest unfolding the 'IC2' acronym, at least in the title. I think IC2A and IC2B should also be explained there before use.

8. In Fig. 2 (and throughout the text) I would probably challenge the expression 'Novel archaeal specificities'. These are 'newly identified' at best; I think the most correct expression would be 'Identified archaeal specificities' or features.

Reviewer #2 (Remarks to the Author):

Short summary.

Suitable Quality? Yes

Results presented are of immediate relevance for many people in your own discipline or for a broader audience? Yes

Conclusions Justified? Not all

Clearly Written? Yes

Procedures Described? Yes

Comments:

Coureux et al. describe a 3.2A cryo-EM structure of the complex of *Pyrococcus abyssi* 30S ribosomal subunit bound with aIF1A, Met-tRNA^{iMet}(A1-U72)-aIF2- GDPNP and mRNA with strong Shine-Dalgarno sequence. The structure uncovers a novel archaeal ribosomal protein aS21, N-terminal extension of L41 and brings insights into base modifications of the rRNA.

This work addresses gaps in information about the atomic details of archaeal small ribosomal subunit structure and our understanding of archaeal translation initiation, as lower-resolution structures of the archaeal translation preinitiation complexes (PIC) did not resolve the network of interactions between initiation factors and the ribosome. Unfortunately, like for many other cryo-EM structures of PICs published, the 3.2A resolution limit applies only to the 30S subunit itself, aIF1A and aIF2 are resolved at much lower resolution, which do not provide information about side-chains positions. Nevertheless, the manuscript also has a number of weaknesses that would have to be revised before it can be considered for publication in CB.

To improve clarity, the following changes are suggested:

1. In Abstract (in the Title too), the authors describe their PIC as a late 30S initiation complex. This definition is not informative enough, better to state (or to add) "30S-aIF1A-aIF2-tRNAⁱ-mRNA complex".

2. Introduction, the authors write: "Previous biochemical and structural studies of a TI complex from *P. abyssi* revealed an active role of aIF2 in start codon selection^{9,10}. Interaction of aIF2 with h44 of the 30S exerts a spring force against accommodation of the initiator tRNA in the P site. When a start codon is found, base pairing with the anticodon compensates for the structural constraint on the TC. This allows a longer stay of the initiator tRNA in the P site and triggers further events, including aIF1 departure and release of aIF2 in its GDP bound form. The dynamic character of the full IC allows local scanning of the initiator tRNA to search for the presence of a start codon in the P site. The dynamic character of the IC, also reflected in toeprinting experiments¹¹, is due to the presence of aIF1 which competes with the initiator tRNA for overlapping binding sites".

Though the active role of the eIF2 in the start codon selection is known/confirmed for decades, the "spring force" and "local scanning" are new terms and should be used with caution. In the ref.#9 (2016 paper from the same lab) the authors hypothesized "that aIF2 exerts a spring force on the tRNA

adjusted in the P site". However, the low resolution of the PICs structures determined in that paper and the lack of supporting experimental data make this mechanism extremely elusive. To my knowledge, no other studies confirmed this mechanism yet. I suggest to change the phrase to: we proposed recently that interaction of aIF2 with h44 of the 30S may exert a spring force#9. Then the mechanism should be described and its hypothetical nature must be stated. Similar concern I have about the term "local scanning". In translation, the "scanning mechanism" is defined as a movement of the PIC along mRNA from 5' to 3' direction to search for the first AUG (Kozak, M., 1978). I believe that the authors describe as a "local scanning" the P-site tRNA transition between the IN and OUT position (rev. in Hinnebusch, A.G, 2017). In PIC described here mRNA does not move and the AUG initiation codon is placed directly in the P site by the interactions between nucleotides of the SD sequence of mRNA and the complimentary sequence of rRNA. If that is correct, then "local scanning" is a wrong term for such process and it is confusing. This part of Introduction should be revised accordingly.

3. Results.

a: Overview of the IC2 cryo-EM structure. "... two relevant conformations of the initiation complex, named IC2A (34k particles, 4.2 Å resolution, Fig. 1a) and IC2B (142k particles, 3.3 Å resolution, Fig. 1b) were identified...". The authors should explain why other conformations of IC are not relevant and were discarded.

b: Identification of novel archaeal specificities of Pab-30S. The authors used 6.6Å resolution structure of the 30S subunit taken from the 70S ribosome (PDB: 4V6U) as a guide but not the higher resolution structure (5.3Å) of the 30S subunit that was determined in their own lab, why?

c: Fig 2: for better/easier identification of aL41 and eS21 on the whole 30S ribosome (Fig2A) - mark them by arrows on the body of a figure. The authors should show (on Fig2 or in the methods) the nucleotide sequence of the new identified ORF for aL41, which database was used etc.)

d: Table 4 - provide references in the table for each published modification included in the table.

e: page 10, top: "A set of rRNA modifications contribute to the stabilization of the codon:anticodon duplex (Fig. 3a and b)..." - mark C34:G3 on the figure 3a,b.

page 10, bottom:

"... identification in the P. abyssi genome of an ortholog of the RsmB enzyme responsible for the corresponding modification in bacteria." - provide reference and the name for the ortholog.

"interaction with the m1A1469-Cm1376 pair" - m6A1469... in the Fig. 3b - please resolve.

page 11, top: "Because the C-terminal part of the aL41 helix interacts with the γ DIII domain of aIF2, this provides a physical link between the P site and the aIF2 γ -h44 contact region" - see comment below for Discussion.

page 12, top: "Following R124, a short α -helix (125-129) leads to the C-terminal part of the protein (Fig. 3a). Faint density for the three terminal residues (130-132) suggests..." - R124 of which protein? C-terminal part of which protein? - not clear from the text and from the Fig.3a. Faint density for the three terminal residues - of which protein? Where is W60 of aIF1A on the Fig.3? - mark these residues on the figure and provide a new figure (supplementary?) with the closer view and the map.

4. Discussion.

First paragraph: "... IC0-PREMOTE and IC1-PIN..." - please provide more details about these ICs here or in Introduction, see also comment 2 about "local scanning". "Consistent with this idea, the IC2 complex, made in the absence of aIF1, shows enhanced stability (present study and11)" - stability of IC was not measured in the study presented.

Second paragraph: "... the conformation of h44 in the vicinity of the binding sites of aIF1 and aIF1A has changed..." - judging from Suppl. fig 11c, similar changes in h44 conformation were observed in ref#16, ref#44, they also might be present in ref#15, #48 (with eIF1 in and out of the P site). The authors should superimpose these structures to see if these rearrangements of h44 are universal. The authors attempt to explain the mechanism of aIF2 dissociation from the 30S subunit after codon-

anticodon base pairing by h44 movement and aIF1 dissociation. It was shown in yeast that release of the phosphate (not GTP hydrolysis itself) allows dissociation of the eIF2 (rev. in Hinnebusch 2011, 2017). Is release of the phosphate needed for aIF2 dissociation from the 30S subunit? If yes, at which step of author's model it should happened? In bacterial, yeast and mammalian PICs the movement of the head of the small ribosomal subunit (rotation, tilting) is also observed upon binding/dissociation of IFs. These head domain movements affect binding of the eIF1, 1A, tRNAⁱ and eIF2. Do the authors observe similar movements of the head of the archaea 30S subunit? If yes, how it affects the binding/conformation of the aIF2- α , β , γ subunits to the ribosome and to the tRNA? page 14, second paragraph: "... This protein could therefore relay some structural information from the P site to the γ DIII binding site and participate in the control of aIF2 release after start codon recognition." The manuscript would be greatly improved if the authors could provide any biochemical data to elucidate under which circumstances aL41 could be separated from the 50S subunit and re-associated with the 30S subunit in vivo. In eukaryotes eL41 is incorporated into ribosomes during the late stage of cytoplasmic pre-60S maturation (MA et al. 2017). Genetic experiments in yeast showed that eL41 is nonessential ribosomal protein and cells lacking eL41 (no eB14 bridge) displayed a similar wild-type growth rate (YU and WARNER 2001; DRESIOS et al. 2003; STEFFEN et al. 2012). However, double mutations severely affect cell growth and demonstrate "a genetic interaction between r-protein eL41 and the eB12 bridge-forming region of eL19 and between r-proteins eL41 and eL24." (Tamm et al., 2019). Taking these into consideration, the authors must be very caution when they suggest that aIL41 remains bound with the 30S subunit in vivo, no experimental data confirm this. Undoubtedly, eL41 is not involved in the PIC formation at the very first initiation event for the newly synthesized 30S subunit. It is very likely that presence of L41 on the small ribosomal subunit is a purification artifact due to the high salt concentration, low magnesium concentration, centrifugation in the gradient of sucrose density etc. In vivo 70S/80S ribosomes are split for subunits by protein factors, which likely leave both subunits intact.

page 15, second paragraph: "...This suggests that the archaeal structure represents a later event in the TI process, before final aIF2 release and binding of aIF5B." There is no eIF5 in archaea, it may be interesting to compare yeast eIF5-PIC with aICs to see structural differences, however, it is rather irrelevant for determination/comparison of the sequence of events in archaea and yeast TI.

page 16, 17 and Fig.4: "... the SD duplex adopted the 'up' tense position in the chamber^{33,36,44} ... ". Ref.#44 - yeast structure. Including yeast structure in the discussion about position of the SD duplex looks irrelevant. In all current structures of eukaryotic PICs, to my knowledge, mRNA in the exit channel does not represent its physiological state, where mRNA is bound to the eIF4F, eIF3 and other proteins involved in cap-dependent translation initiation and scanning. It would be too early to conclude that mRNA is in "UP" position in eukaryotic PIC in vivo.

5. Methods. Page 19, top: "... Because very weak electron density for aIF1A and aIF2 was observed in IC2C, this conformation was not considered as an interesting functional state...". I suggest to change it to something like: we did not further refine the IC2C because of the very weak electron density for aIF1A and aIF2.

6. Some references (#25, 49, 79, 80, 89) do not have complete information - please correct.

Reviewer #3 (Remarks to the Author):

In this study, Coureux and colleagues report the first high-resolution Cryo-EM structure of a late archaeal translation initiation complex. This study is very interesting from various perspectives.

The main findings are:

i) the first high resolution structure of an archaeal small ribosomal subunit,

- ii) structural basis for the archaeal translation initiation cycle, that allows comparative structural analysis of translation initiation across the tree of life,
- iii) new insights into "archaeal" ribosomal subunit features (rRNA modifications/ribosomal proteins).

Overall, it is a beautiful piece of work that provides significant and appealing insights to our general understanding of ribosome biology.

In order to improve some aspects of the manuscript I have some suggestions/comments that should be addressed.

Specific points:

1) "novel archaeal specificities": the authors describe the presence of large amounts of acetylated cytidine scattered across the 16S rRNA, or N-terminal extension of aL41, a new r-protein aS21, which are described as novel archaeal features (e.g. title of result section, figure title ...). The scientific/experimental basis to make a very general statement is not provided (see also below) and is very likely to be wrong (at least to some extent), e.g. widespread of acetylated cytidine in non-hyperthermophile archaea?! I have no problem on speculation regarding these aspects but I think the authors should be more cautious when presenting these findings in the result section and discuss their putative implications more broadly later on.

2) Ribosomal proteins novelty.

The authors describe the un-annotated N-terminal extension of *Pyrococcus abyssi* aL41. To better appreciate the conservation/distribution of this feature, it would be useful to provide sequence alignments of representative archaeal aL41. This r-protein is in principle a large ribosomal subunit r-protein, and it is somehow surprising to find it in the small ribosomal subunit context. Previous work have described promiscuous behavior of some r-proteins in archaea (PMID: 23222135), but I am not sure that it really applies for this particular case. In any case it would be good for the readers to shortly clarify/comment on this.

The authors describe a new (archaeal specific) ribosomal protein aS21. Here again the distribution and sequence conservation of this new r-protein across archaea (and beyond) is not clear and should be clarified.

Please provide database entry of *P. abyssi* aS21 and more detail sequence alignments.

As a side note, the authors should also make sure that these findings are integrated in relevant public databases.

3) Ribosomal RNA acetylation.

The authors speculate on the possible role of acetylation for (hyper-) thermostability. Remarkably, *P. abyssi* can be cultivated across a rather wide range of temperature (67-102°C !!!) (<https://doi.org/10.1007/BF00252219>), it would be interesting to analyze acetylation dynamic at different temperatures.

The authors proposed Nat10/Kre33 as putative acetyltransferase and proposed a sequence recognition motif (CCG) as "systematic" modification target (e.g. end of discussion). However, there are apparently also many CCG motifs which are not modified (e.g. h1, h6...)? The presence of this motif does not seem to be sufficient (or are these only modified at higher temperature, see above comment).

There are different studies (e.g. PMID: 28542199, PMID: 25653167) suggesting that Nat10/Kre33 targeting is mediated by accessory factors (protein or snoRNA ...). Moreover, in these organisms Nat10/Kre33 activity seems to be better restricted/controlled, whereas the apparent distribution of the acetylation in *P. abyssi* may suggest a rather promiscuous or unknown mechanism. In those

lines how well conserved are the modified residues? I think there is room to broaden and deepen the discussion.

Finally, some "acetylated" residues could not be verified by chemical derivation. Can the authors comment on that?

4) hm5C modification at position C1378.

The equivalent position is known to be modified in some archaea and *T. maritima* (PMID: 18844986 and references therein). This unknown modification is described as N330 (according to its mass in Da). Accordingly, the nature of this modification seems to vary in archaea. On the other hand, the authors do not seem to be very confident with this assignment as well.

5) Citations:

In several occurrence appropriate citation(s) should be added to support the claim(s).

Few examples are indicated below.

- Introduction §2 first sentence "In archaea, ... 5' UTR." Reference(s)?

- Introduction §2 third sentence "Moreover, ... two ribosomes." Reference(s)?

... more of these are scattered throughout the manuscript.

6) Results §2: "... KsgA/Dim1 ... is conserved throughout evolution". This is not completely true since at least in *Nanoarchaeum equitans* KsgA/Dim1 is not found, and 2' O methylations of h45 occur instead of dimethylation of the two universally conserved adenosines (PMID: 28204608).

7) eS26 (Discussion – The archaeal mRNA exit chamber vs eukaryotic and bacterial ones)

The authors discuss differences between archaeal/eukaryotic mRNA exit channels. Among the discussed differences is ribosomal protein eS26. However, eS26 is not lacking in all archaea, but is clearly present in Crenarchaeota/Lokiarchaeota but absent from Euryarchaeota (e.g. PMID: 12490706 or PMID: 30201955). From this point of view archaea have heterogeneous archaeal mRNA exit chambers, and the current comparison can at best applied to Euryarchaeota vs Eukarya.

8) Title/keywords list.

The title maybe too specific and does not fully reflect ALL the new insights of this study.

Maybe some additional specific keywords should also be incorporated to ensure appropriate visibility of the work.

Point-by-point responses to the reviews COMMSBIO-19-1340-T

Former title: Cryo-EM study of a late 30S archaeal translation initiation complex: insights into evolution of molecular mechanisms

New title : The 30S:mRNA:aIF1A:aIF2:GTP:Met-tRNA_i^{Met} cryo-EM structure reveals archaeal features and gives insights into evolution of translation initiation.

We thank the reviewers for their positive and meticulous assessments of our work. Their detailed suggestions were very helpful to improve the clarity and impact of our manuscript.

We provide below our point-by-point responses to the comments, which detail our changes to the manuscript. To provide a quick overview of our revisions, we have highlighted in red all new or altered portions of text in the revised manuscript files.

REVIEWER 1:

Manuscript by Pierre-Damien Coureux *et alia* describes electron density maps inferred by using cryogenic transmission electron microscopy imaging of purified archaeal translation initiation complexes. The electron density maps are refined to high resolutions and classified to clearly segregate into several classes. Two major classes and the corresponding initiation complex conformations (or complex types) were detectable, one with fully and another with partially accommodated methionylated initiator tRNA. The high resolution obtained in the study allowed attribution of novel feature types within the structures, such as species-specific modifications of the ribosomal RNA.

It is an excellent study. The work significantly adds to the current knowledge. While the existing PDB entries 5JB3 and 5JBH (from the same group; released in 2016) of similar archaeal translation initiation complexes outlined the overall layout of these structures, the newly refined data provides much higher resolution (~3.3 vs. ~ 5.3 Å). The refinement parameters are also stricter in the current manuscript. This allows for the visualisation of smaller features and adds much certainty to the structures and modelling. The manuscript is clearly written; delivery of the results is on a very high level; the flow of the text is concise and logically well-ordered. I have no doubts the work will be cited and will become an excellent resource.

I have no major criticisms or objections of any kind. I have a few technical comments or suggestions; it would be nice to see these implemented before acceptance. I can highly recommend this work for publication in Communications Biology generally in the current form.

We thank reviewer 1 for these very positive comments and for the helpful suggestions.

Major Point

1. Did the authors attempt classification *ab initio* and what was the actual reason for dropping it for the initial refinement? I also think that not enough attention was diverted to the classification. What the other types of complexes could represent? I understand that this may be the focus of subsequent studies, or the authors deemed the other types as 'uninteresting'. However, I think that the other classes of the complexes identified should be described with more detail.

At the time the data processing was done in the Relion 2.1 workflow suite, the SGD (Stochastic Gradient Descent) *ab initio* reconstruction was not implemented yet. We used the 30S subunit from *P. furiosus* filtered at 60 Å resolution as an initial model (as in Coureux et al, 2016) and converged

straightforwardly to high resolution (3.3Å). A first round of 3D classification was then performed to improve electron density for aIF2, aIF1A and tRNA, and to discard classes that did not show a nice 30S density. Several subsequent rounds of classification were attempted to identify the different conformations of the IC2 complex. The strategy that gave the best results is presented in Supplementary Figure 1. The "IC2C" class contains very weak density for aIF1A and was therefore not considered representative of an IC2 conformation. This is the reason why this class has not been further refined. We have now better explained this choice in the results section (bottom of p5 and beginning of p6) and in the Materials and Methods section (p20). In Supplementary Fig. 1, we clarified the reason why some classes were discarded following 3D classification by changing "poorly resolved" into "30S poorly defined" in the flowchart. Moreover, following the remark of the reviewer, we have also tried classification using initial models generated de novo by the SGD method in Relion 3.0.5. This strategy did not reveal conformations that were not identified before. Rather, classification was less efficient with more particles in bad classes showing poorly defined 30S.

Minor points

Abstract

1. While hyphenation for the description of the components of molecular complexes is acceptable (and it is a widespread practice), in my opinion, the use of colon or dot signs for the non-covalent interactions is better, as it discriminates covalent bonds or interactions from the non-covalent counterparts. For example, in the variant 'aIF2-GTP-Met-tRNA_i^{Met}' used by the authors already in the abstract, it seems that the methionine attachment to tRNA is of the same nature as the interactions with the other components. For a good structural study, replacing this with aIF2:GTP:Met-tRNA_i^{Met} or aIF2•GTP•Met-tRNA_i^{Met} (depending on the style preferences) could be a desirable improvement.

Following the reviewer's recommendation, we have now used the colon sign to symbolize the non-covalent aIF2:GTP:Met-tRNA_i^{Met} ternary complex throughout the text.

Introduction

2. In the introduction (and other parts), I suggest replacing methionyl-initiator tRNA with methionylated initiator tRNA.

The text has been corrected accordingly.

3. I suggest explicitly specifying 'assist the selection mechanism' -> 'assist the start codon selection mechanism'.

The text has been corrected accordingly.

4. Inconsistent five or three prime sign use (sometimes apostrophe, sometimes prime sign).

The prime sign is now used throughout the text.

5. I am unsure if the 'spring force' is the best wording; is it purely an elastic deformation (changes in the bonds lengths/angles), or there is an entropic component as well in this case (mobility restriction)?

In a previous study (Coureux et al, 2016), we observed that when tRNA was not accommodated in the P site, the conformation of the TC was identical to that observed in solution. However, when tRNA was engaged within the P site, the structure of the TC deviated from the structure in solution. We concluded that simultaneous tRNA accommodation in the P site and maintenance of the contact of aIF2 γ with h44 caused a structural strain in the ternary complex. Because the TC should tend to adopt its most stable position, with the lowest free energy, we hypothesized that aIF2 should

exert a “restoring force” on tRNA_i counteracting its accommodation in the P site, because of the energetic penalty. Codon-anticodon pairing would then partly compensate for the restoring force, allowing a longer stay of the tRNA in the P site if a start codon is present. We agree with the reviewer that the penalty has both enthalpic and entropic components. Because the wording “spring force” seems confusing, we now better explain the model developed in our previous study and use the wording “restoring force” instead. The wording “spring force” is now employed only once just to make a link with the previous study (see Introduction page 4 first paragraph).

Results

6. I would probably suggest using ‘ π - π interactions’ instead of ‘pi interactions’.
The text has been corrected accordingly.

Figure Legends

7. In Fig. 1 legend, I suggest unfolding the ‘IC2’ acronym, at least in the title. I think IC2A and IC2B should also be explained there before use.

According to the reviewer's remark, we changed Fig. 1 legend as follows:

Fig. 1: Cryo-EM maps of two conformations of the initiation complex 30S:mRNA:aIF1A:TC (IC2). The figure describes the two refined conformations, IC2A and IC2B, resulting from 3D classification. These two conformations essentially differ by the position of aIF2 with respect to h44 (see text)...

8. In Fig. 2 (and throughout the text) I would probably challenge the expression ‘Novel archaeal specificities’. These are ‘newly identified’ at best; I think the most correct expression would be ‘Identified archaeal specificities’ or features.

We changed ‘Novel archaeal specificities’ by ‘Identified archaeal specificities of Pa-30S’ as suggested by the reviewer in the title of Fig. 2 and throughout the text. As a result the term 'Novel' is no longer used.

REVIEWER 2:

Coureau et al. describe a 3.2Å cryo-EM structure of the complex of *Pyrococcus abyssi* 30S ribosomal subunit bound with aIF1A, Met-tRNA_iMet(A1-U72)-aIF2- GDPNP and mRNA with strong Shine-Dalgarno sequence. The structure uncovers a novel archaeal ribosomal protein aS21, N-terminal extension of L41 and brings insights into base modifications of the rRNA.

This work addresses gaps in information about the atomic details of archaeal small ribosomal subunit structure and our understanding of archaeal translation initiation, as lower-resolution structures of the archaeal translation preinitiation complexes (PIC) did not resolve the network of interactions between initiation factors and the ribosome. Unfortunately, like for many other cryo-EM structures of PICs published, the 3.2Å resolution limit applies only to the 30S subunit itself, aIF1A and aIF2 are resolved at much lower resolution, which do not provide information about side-chains positions. Nevertheless, the manuscript also has a number of weaknesses that would have to be revised before it can be considered for publication in CB.

We thank reviewer 2 for his positive and thorough feedback on our work.

1. In Abstract (in the Title too), the authors describe their PIC as a late 30S initiation complex. This definition is not informative enough, better to state (or to add) “30S-aIF1A-aIF2-tRNA_i-mRNA complex”.

The abstract and the title have been modified according to the recommendation of the reviewer (See also reviewer 3 point 8).

The new title is : The 30S:mRNA:aIF1A:aIF2:GTP:Met-tRNA_i^{Met} cryo-EM structure reveals archaeal features and gives insights into evolution of translation initiation.

2. Introduction, the authors write: “Previous biochemical and structural studies of a TI complex from *P. abyssi* revealed an active role of aIF2 in start codon selection^{9,10}. Interaction of aIF2 with h44 of the 30S exerts a spring force against accommodation of the initiator tRNA in the P site. When a start codon is found, base pairing with the anticodon compensates for the structural constraint on the TC. This allows a longer stay of the initiator tRNA in the P site and triggers further events, including aIF1 departure and release of aIF2 in its GDP bound form. The dynamic character of the full IC allows local scanning of the initiator tRNA to search for the presence of a start codon in the P site. The dynamic character of the IC, also reflected in toeprinting experiments¹¹, is due to the presence of aIF1 which competes with the initiator tRNA for overlapping binding sites”.

Though the active role of the eIF2 in the start codon selection is known/confirmed for decades, the “spring force” and “local scanning” are new terms and should be used with caution. In the ref.#9 (2016 paper from the same lab) the authors hypothesized “that aIF2 exerts a spring force on the tRNA adjusted in the P site”. However, the low resolution of the PICs structures determined in that paper and the lack of supporting experimental data make this mechanism extremely elusive. To my knowledge, no other studies confirmed this mechanism yet. I suggest to change the phrase to: we proposed recently that interaction of aIF2 with h44 of the 30S may exert a spring force#9. Then the mechanism should be described and its hypothetical nature must be stated. Similar concern I have about the term “local scanning”. In translation, the “scanning mechanism” is defined as a movement of the PIC along mRNA from 5’ to 3’ direction to search for the first AUG (Kozak, M., 1978). I believe that the authors describe as a “local scanning” the P-site tRNA transition between the IN and OUT position (rev. in Hinnebusch, A.G, 2017). In PIC described here mRNA does not move and the AUG initiation codon is placed directly in the P site by the interactions between nucleotides of the SD sequence of mRNA and the complimentary sequence of rRNA. If that is correct, then “local scanning” is a wrong term for such process and it is confusing. This part of Introduction should be revised accordingly.

As recommended by the reviewer, the hypothetical nature of the "spring force" model is now clearly stated (p4, lines 8-20). See also point 5 of reviewer 1.

Our view of "local scanning" during translation initiation in archaea is as follows. Interaction of the SD sequence with 16S rRNA (or of the IC with the 5'-end of mRNA if no SD is present) allows pre-positioning of the IC in the vicinity of the start codon. However, this is not sufficient to place the start codon exactly in the P site. Therefore, after pre-positioning, the IC can move on the mRNA (*i.e.* scan it locally) in order to search for a start codon precisely located in the P site. The start codon is then identified through base pairing with the anticodon of the initiator tRNA, as it is the case when eukaryotic long-range scanning stops. Here, because we used a model mRNA (corresponding to the highly translated aEF1-A mRNA) where the SD sequence is likely to be optimally positioned with respect to the start codon, the IC is probably pre-positioned with the start codon close or even in the P site. However, previous toeprinting studies reported in Monestier et al (2018) have illustrated the displacement of the IC induced by start codon-tRNA anticodon base pairing in mRNAs with non-optimally positioned SD sequences. We agree with the reviewer that our description was too short in the first version and have now added a more precise explanation (bottom of page 3-top of page 4).

3. Results.

a: Overview of the IC2 cryo-EM structure. “... two relevant conformations of the initiation complex, named IC2A (34k particles, 4.2 Å resolution, Fig. 1a) and IC2B (142k particles, 3.3 Å resolution, Fig.

1b) were identified...". The authors should explain why other conformations of IC are not relevant and were discarded.

The "IC2C" class contains very weak density for aIF1A and was therefore not considered representative of an IC2 conformation. This is the reason why this class has not been further refined. We have now better explained this choice in the results section (bottom of p5 and beginning of p6) and in the Materials and Methods section (p20). In Supplementary Fig. 1, we clarified the reason why some classes were discarded following 3D classification. To this aim, we changed "poorly resolved" by "30S poorly defined" in the flowchart (see also reviewer 1 major point).

b: Identification of novel archaeal specificities of Pab-30S. The authors used 6.6Å resolution structure of the 30S subunit taken from the 70S ribosome (PDB: 4V6U) as a guide but not the higher resolution structure (5.3Å) of the 30S subunit that was determined in their own lab, why?

We cited the 6.6 Å resolution structure of the 30S subunit taken from the 70S ribosome (PDB: 4V6U) and not the 5JB3 structure because the 5JB3 model essentially corresponds to a fit of 4V6U in the electron density of 5JB3. In 5JB3, because the resolution was only 5.3 Å, it was not possible to construct a full atomic model for the *P. abyssi* 30S and the sequences of rRNA and r-proteins from *P. furiosus* were kept. This is described in the Methods section of Coureux, et al., 2016. Citing the 4V6U structure in the text seemed fairer to us to give proper credit to the original paper by Armache et al., 2013. In the revised version both 4V6U and 5JB3 are mentioned in Table 1 and in the Methods section (page 21 line 15) as well as 2AHO, 4RD4, 2QMU, 3V11, 2MNO that were used to model the TC and aIF1A (Table 1 and page 22 lines 3 and 7). We apologize for this lack of precision, which has now been corrected.

c: Fig 2: for better/easier identification of aL41 and eS21 on the whole 30S ribosome (Fig2A) - mark them by arrows on the body of a figure. The authors should show (on Fig2 or in the methods) the nucleotide sequence of the new identified ORF for aL41, which database was used etc.)

We added a full view of the 30S in which locations of aS21 and aL41 are indicated (Supplementary Fig. 7a right panel, new numbering). We preferred not to use Fig2a for this purpose because aS21 is not visible in the orientation chosen to illustrate the overall IC2B structure. The nucleotide sequence of the new identified ORF for aL41 is now indicated in Supplementary Fig. 6b as well as sequence alignment of putative aL41 from archaeal genomes, including accession numbers (Supplementary Fig.6c). The NCBI reference for the nucleotidic sequence of Pa-aL41 is indicated in the legend of Supplementary Fig. 6.

d: Table 4 - provide references in the table for each published modification included in the table.

References used for Table 4 are now indicated in the legend.

e: page 10, top: "A set of rRNA modifications contribute to the stabilization of the codon:anticodon duplex (Fig. 3a and b)..." - mark C34:G3 on the figure 3a,b.

The C34:G3 bases are now marked in Figure 3b and G3 in Figure 3a.

page 10, bottom:

"... identification in the *P. abyssi* genome of an ortholog of the RsmB enzyme responsible for the corresponding modification in bacteria." - provide reference and the name for the ortholog.

The accession number of the ortholog is now provided. We also added a reference describing the structural characterization of the *Pyrococcus horikoshii* ortholog (Page 11 lines 7 and 8).

“interaction with the m1A1469-Cm1376 pair” - m6A1469... in the Fig. 3b - please resolve.

The text has been corrected (page 11 line 8 from bottom). The modification is m⁶A1469. We apologize for this error.

page 11, top: “Because the C-terminal part of the α L41 helix interacts with the γ DIII domain of aIF2, this provides a physical link between the P site and the aIF2 γ -h44 contact region” - see comment below for Discussion.

This point is addressed below.

page 12, top: “Following R124, a short α -helix (125-129) leads to the C-terminal part of the protein (Fig. 3a). Faint density for the three terminal residues (130-132) suggests...” - R124 of which protein? C-terminal part of which protein? - not clear from the text and from the Fig.3a. Faint density for the three terminal residues - of which protein? Where is W60 of aIF1A on the Fig.3? - mark these residues on the figure and provide a new figure (supplementary?) with the closer view and the map.

The paragraph has been corrected to clearly mention that the description refers to uS19. A close view of these residues including the map was shown in Supplementary Fig. 6c (now 7c), but was not appropriately cited in the text. This is now done (end of the results section, bottom of page 12, top of page 13). W60 of aIF1A is now drawn and labeled in Fig. 3a and uS19-R124 is labeled in Fig. 3b, as suggested by the reviewer.

4. Discussion.

First paragraph: “... IC0-PREMOTE and IC1-PIN...” - please provide more details about these ICs here or in Introduction, see also comment 2 about “local scanning”. “Consistent with this idea, the IC2 complex, made in the absence of aIF1, shows enhanced stability (present study and11)” - stability of IC was not measured in the study presented.

Introduction and the first paragraph of the discussion have been re-written to better explain the nature of IC0-P_{REMOTE} and IC1-P_{IN} (See also response to point 2 and to reviewer 1 point 5).

The text has also been corrected to state that this enhanced stability is deduced from toeprinting experiments described in ref.13 (page 13, end of the first paragraph).

Second paragraph: “... the conformation of h44 in the vicinity of the binding sites of aIF1 and aIF1A has changed...” - judging from Suppl. fig 11c, similar changes in h44 conformation were observed in ref#16, ref#44, they also might be present in ref#15, #48 (with eIF1 in and out of the P site). The authors should superimpose these structures to see if these rearrangements of h44 are universal.

In ref#16, the eukaryotic PIC complex contained eIF5 bound at the position vacated by eIF1 departure. In ref#44, a bacterial IC is described with IF3 bound close to the P site. In ref#15, eIF1 is always present in a eukaryotic PIC. In ref#48, eIF1 is bound but not eIF2. Therefore, comparison of these structures with the present one is very difficult to interpret. In order to avoid over-interpretation, we preferred not to discuss the possible universality of the h44 rearrangements observed here. Rather, taking into account the reviewer's remark, we added a sentence to note that movements of h44 related to eIF1 binding have already been observed in eukaryotes (page 15, lines 6-8, Lomakin and Steitz 2013 and Weisser, et al., 2013).

The authors attempt to explain the mechanism of aIF2 dissociation from the 30S subunit after codon-anticodon base pairing by h44 movement and aIF1 dissociation. It was shown in yeast that release of

the phosphate (not GTP hydrolysis itself) allows dissociation of the eIF2 (rev. in Hinnebusch 2011, 2017). Is release of the phosphate needed for aIF2 dissociation from the 30S subunit? If yes, at which step of author's model it should happened?

We totally agree with the reviewer that Pi release is a key event in aIF2 release. This point is now more clearly addressed in the revised version of the discussion and in the legend of Figure 5. The sequence of events has been clarified and relevant publications cited (page 14, lines 5-12).

In bacterial, yeast and mammalian PICs the movement of the head of the small ribosomal subunit (rotation, tilting) is also observed upon binding/dissociation of IFs. These head domain movements affect binding of the eIF1, 1A, tRNAⁱ and eIF2. Do the authors observe similar movements of the head of the archaea 30S subunit? If yes, how it affects the binding/conformation of the aIF2-alfa, beta, gamma subunits to the ribosome and to the tRNA?

Similar head motions were observed between IC0 and IC1 states and have been described in Coureux et al., 2016. The head motions are now reminded in the first paragraph of the discussion (p13). In contrast, in the present IC2 structure, we did not observe significant head motion as compared to IC1. This is now more clearly stated in the results p6 line 4.

page 14, second paragraph: "... This protein could therefore relay some structural information from the P site to the γ DIII binding site and participate in the control of aIF2 release after start codon recognition." The manuscript would be greatly improved if the authors could provide any biochemical data to elucidate under which circumstances aL41 could be separated from the 50S subunit and re-associated with the 30S subunit in vivo. In eukaryotes eL41 is incorporated into ribosomes during the late stage of cytoplasmic pre-60S maturation (MA et al. 2017). Genetic experiments in yeast showed that eL41 is nonessential ribosomal protein and cells lacking eL41 (no eB14 bridge) displayed a similar wild-type growth rate (YU and WARNER 2001; DRESIOS et al. 2003; STEFFEN et al. 2012). However, double mutations severely affect cell growth and demonstrate "a genetic interaction between r-protein eL41 and the eB12 bridge-forming region of eL19 and between r-proteins eL41 and eL24."

(Tamm et al., 2019). Taking these into consideration, the authors must be very caution when they suggest that aL41 remains bound with the 30S subunit in vivo, no experimental data confirm this. Undoubtedly, eL41 is not involved in the PIC formation at the very first initiation event for the newly synthesized 30S subunit. It is very likely that presence of L41 on the small ribosomal subunit is a purification artifact due to the high salt concentration, low magnesium concentration, centrifugation in the gradient of sucrose density etc. In vivo 70S/80S ribosomes are split for subunits by protein factors, which likely leave both subunits intact.

To our knowledge, yeast L41 protein was identified and named by Otaka et al., Mol Gen Genet. 1984 after characterization of protein content of ribosomal subunits separated by sucrose gradient (5-32% (w/w) in 50 mM Tris-HCl, pH 7.4, 0.1 mM MgCl₂, and 1 mM DTT; Higo and Otaka Biochemistry, 1979).

In 2011, Ben-Shem et al., Science, described the first crystallographic structure of *S. cerevisiae* 80S structure. They commented, "*The bridge is formed by protein L41e, which consists of a single α helix that is enveloped by conserved core rRNA. L41e protrudes from 60S into a binding pocket in the small subunit, which is lined by helices h27, h45, and h44, in proximity to the decoding center. Curiously, in the context of the full ribosome, L41e is much more strongly associated with 40S than with 60S.*"

Since the Ben-Shem et al structure, several medium and high-resolution structures of small ribosomal subunits from eukaryotes (3JAP, 6FYX, 6P4H...) and archaea (5JB3) include L41 consistent with the first observation in the yeast 80S.

The references cited by the reviewer (YU and WARNER 2001; DRESIOS et al. 2003; STEFFEN et al.

2012; Tamm et al., 2019) do not seem to contradict this view.

In Ma et al, 2017, the structure of the pre-60S ribosomal particle described do not contain eL41 and the authors commented, "In terms of ribosomal proteins, uL16, uL10, uL11, eL40 and eL41 were clearly missing from the Nmd3 particles (Supplementary Fig. 3a,b). eL41, as a single α -helical protein, binds at the subunit interface, with a stronger association with the 40S subunit²⁷. Therefore, its absence is not surprising".

Overall, the data seem to indicate that L41, involved in a bridge between the large and the small ribosomal subunit is more strongly bound to the small ribosomal subunit. This is now illustrated in Supplementary Fig. 6a.

We agree that we should have mentioned in the manuscript that the main binding site of e/aL41 is on the small subunit, in contrast to what would be expected according to its name. This is now stated in the results section (p7, lines 2-6) with references to the original Ben-Shem et al. 2011 paper and to eukaryotic 40S structures. Moreover, according to the reviewer's recommendation, we now more clearly state that a possible role of aL41 in the TI mechanism is only a hypothesis coming from our structural observations (Discussion page 15 line 2 from bottom).

page 15, second paragraph: "...This suggests that the archaeal structure represents a later event in the TI process, before final aIF2 release and binding of aIF5B." There is no eIF5 in archaea, it may be interesting to compare yeast eIF5-PIC with aICs to see structural differences, however, it is rather irrelevant for determination/comparison of the sequence of events in archaea and yeast TI.

We removed the sentence cited by the reviewer. We now mention that a step following the one illustrated by the KI-PIC structure (Llacer et al, 2018, Elife) may be a repositioning of eIF1A and a relocation of the C-terminal tails of uS13 and uS19.

page 16, 17 and Fig.4: "... the SD duplex adopted the 'up' tense position in the chamber^{33,36,44} ... ". Ref.#44 - yeast structure. Including yeast structure in the discussion about position of the SD duplex looks irrelevant. In all current structures of eukaryotic PICs, to my knowledge, mRNA in the exit channel does not represent its physiological state, where mRNA is bound to the eIF4F, eIF3 and other proteins involved in cap-dependent translation initiation and scanning. It would be too early to conclude that mRNA is in "UP" position in eukaryotic PIC in vivo.

The discussion concerning the "UP" position indeed does not concern the eukaryotic case. Ref#44 describes bacterial complexes (*T. thermophilus*) from the Ramakrishnan group (Hussain et al., Cell 2016) not yeast ones.

5. Methods. Page 19, top: "... Because very weak electron density for aIF1A and aIF2 was observed in IC2C, this conformation was not considered as an interesting functional state...". I suggest to change it to something like: we did not further refine the IC2C because of the very weak electron density for aIF1A and aIF2.

This has been done, see page 21 lines 13-14 and also reviewer 1 major point.

6. Some references (#25, 49, 79, 80, 89) do not have complete information - please correct.

We apologize for these errors. The references have been checked and corrected.

REVIEWER 3:

In this study, Coureux and colleagues report the first high-resolution Cryo-EM structure of a late archaeal translation initiation complex. This study is very interesting from various perspectives.

The main findings are:

- i) the first high resolution structure of an archaeal small ribosomal subunit,
- ii) structural basis for the archaeal translation initiation cycle, that allows comparative structural analysis of translation initiation across the tree of life,
- iii) new insights into “archaeal” ribosomal subunit features (rRNA modifications/ribosomal proteins).

Overall, it is a beautiful piece of work that provides significant and appealing insights to our general understanding of ribosome biology.

In order to improve some aspects of the manuscript I have some suggestions/comments that should be addressed.

We thank reviewer 3 for these very positive comments and for the helpful suggestions.

1) “novel archaeal specificities”: the authors describe the presence of large amounts of acetylated cytidine scattered across the 16S rRNA, or N-terminal extension of aeL41, a new r-protein aS21, which are described as novel archaeal features (e.g. title of result section, figure title ...). The scientific/experimental basis to make a very general statement is not provided (see also below) and is very likely to be wrong (at least to some extent), e.g. widespread of acetylated cytidine in non-hyperthermophile archaea?! I have no problem on speculation regarding these aspects but I think the authors should be more cautious when presenting these findings in the result section and discuss their putative implications more broadly later on.

The title “identification of novel archaeal specificities of Pa-30S” in the results section has been changed by “Identification of archaeal specificities of Pa-30S ” and the title of Figure 2 is now “archaeal specificities of Pa-30S”.

Moreover, in order to reinforce the present structural analysis, additional information is now given. A sequence alignment of aL41 deduced from blast search using aL41 from *Pyrococcus abyssi* as a query is now provided in a new Supplementary Fig. 6. This search revealed aL41 in genomes of some Thermococcales and DPANN. This is consistent with previous work (cited in the text; ref8) suggesting that aL41 is not found in all archaeal genomes.

A sequence alignment of aS21 is now provided showing that the protein is widespread in archaea (Supplementary Data 1, cited in page 7 line 1 from bottom).

Finally, we agree with the reviewer that N⁴-acetyl-cytidines are likely to be less present in non-hyperthermophilic organisms. Please see response to point 3 below.

2) Ribosomal proteins novelty.

The authors describe the un-annotated N-terminal extension of *Pyrococcus abyssi* aeL41. To better appreciate the conservation/distribution of this feature, it would be useful to provide sequence alignments of representative archaeal aeL41. This r-protein is in principle a large ribosomal subunit r-protein, and it is somehow surprising to find it in the small ribosomal subunit context. Previous work have described promiscuous behavior of some r-proteins in archaea (PMID: 23222135), but I am not sure that it really applies for this particular case. In any case it would be good for the readers to

shortly clarify/comment on this.

See point 1 above for alignments of representative archaeal aeL41.

Moreover, a sequence alignment of aL41 and eL41 from *S. cerevisiae* is now shown in Figure 2c.

For the binding site of eaL41 on the small ribosomal subunit, see response to reviewer 2 discussion, point 4, results section page 7, lines 2-5 and Supplementary Fig. 6a.

The authors describe a new (archaeal specific) ribosomal protein aS21. Here again the distribution and sequence conservation of this new r-protein across archaea (and beyond) is not clear and should be clarified.

Please provide database entry of *P. abyssi* aS21 and more detail sequence alignments.

As a side note, the authors should also make sure that these findings are integrated in relevant public databases.

A sequence alignment of aS21 is now provided showing that the protein is widespread in archaea (Supplementary Data 1 cited in page 7, line 1 from bottom).

The accession number of the aS21 sequence is now indicated in the text page 7, line 8 from bottom.

3) Ribosomal RNA acetylation.

The authors speculate on the possible role of acetylation for (hyper-) thermostability. Remarkably, *P. abyssi* can be cultivated across a rather wide range of temperature (67-102°C !!!) (<https://doi.org/10.1007/BF00252219>), it would be interesting to analyze acetylation dynamic at different temperatures.

The authors proposed Nat10/Kre33 as putative acetyltransferase and proposed a sequence recognition motif (CCG) as “systematic” modification target (e.g. end of discussion). However, there are apparently also many CCG motifs which are not modified (e.g. h1, h6...)? The presence of this motif does not seem to be sufficient (or are these only modified at higher temperature, see above comment).

There are different studies (e.g. PMID: 28542199, PMID: 25653167) suggesting that Nat10/Kre33 targeting is mediated by accessory factors (protein or snoRNA ...). Moreover, in these organisms Nat10/Kre33 activity seems to be better restricted/controlled, whereas the apparent distribution of the acetylation in *P. abyssi* may suggest a rather promiscuous or unknown mechanism. In those lines how well conserved are the modified residues? I think there is room to broaden and deepen the discussion.

We agree with the reviewer that it would be very interesting to study the dynamics of the acetylation profile at various temperatures and we thank the reviewer for this hypothesis. This is a heavy study, especially because *P. abyssi* is a strict anaerobe and because quantitative measurements of N4-acetylation of so many cytidines are not straightforward. Such a study is therefore beyond the scope of the present manuscript. We hope that our study will stimulate research in this field. N4-acetylation in archaeal ribosomal rRNA clearly deserves further investigation. As discussed by the reviewer, the mechanism of N4-acetylation in archaeal ribosomes is still unknown. As mentioned, Nat10/Kre33 orthologue would be a good candidate. Regarding the wide distribution of N4-acetylations and the non-systematic modification of all CCG sequences, small RNAs would likely assist the Nat10 orthologue. These considerations have now been incorporated in the text, see Results section page 9 lines 6-10 from bottom.

Finally, some “acetylated” residues could not be verified by chemical derivation. Can the authors comment on that?

In 3 cases, we did not observe an RT stop at the expected position. This may be due to lower reactivity towards sodium borohydride reduction. This is now stated in Table 2 legend. The electron density was however non ambiguous for the three positions (see Supplementary Figure 8).

4) hm5C modification at position C1378.

The equivalent position is known to be modified in some archaea and *T. maritima* (PMID: 18844986 and references therein). This unknown modification is described as N330 (according to its mass in Da). Accordingly, the nature of this modification seems to vary in archaea. On the other hand, the authors do not seem to be very confident with this assignment as well.

In agreement with the reviewer's comment we mention in the text (page 8 line 11 from bottom) that the hm5C modification is only tentative. Moreover, we added a remark in the legend of Table 3 to mention that the corresponding position is known to carry an unknown modification in some archaea. The review PMID: 18844986 is now cited.

5) Citations:

In several occurrence appropriate citation(s) should be added to support the claim(s).

Few examples are indicated below.

- Introduction §2 first sentence "In archaea, ... 5'UTR." Reference(s)?

- Introduction §2 third sentence "Moreover, ... two ribosomes." Reference(s)?

... more of these are scattered throughout the manuscript.

We have added appropriate citations for the claims quoted by the reviewer as well as at several other places in the revised manuscript. Unfortunately, we are limited by the editorial requirement (Articles allow in principle up to 70 references). The present number of references is 98. We apologize for all the studies that could not be cited.

6) Results §2: "... KsgA/Dim1 ... is conserved throughout evolution". This is not completely true since at least in *Nanoarchaeum equitans* KsgA/Dim1 is not found, and 2'O methylations of h45 occur instead of dimethylation of the two universally conserved adenosines (PMID: 28204608).

We changed the sentence and cited the corresponding reference p8, line 16. We thank the reviewer for this information.

7) eS26 (Discussion – The archaeal mRNA exit chamber vs eukaryotic and bacterial ones)

The authors discuss differences between archaeal/eukaryotic mRNA exit channels. Among the discussed differences is ribosomal protein eS26. However, eS26 is not lacking in all archaea, but is clearly present in Crenarchaeota/Lokiarchaeota but absent from Euryarchaeota (e.g. PMID: 12490706 or PMID: 30201955). From this point of view archaea have heterogeneous archaeal mRNA exit chambers, and the current comparison can at best applied to Euryarchaeota vs Eukarya.

We fully agree with the reviewer that according to the conservation of aS26 in archaeal genomes, the discussion concerning the differences between archaeal/eukaryotic mRNA exit channels has to be restricted to the case of euryarchaeota. Therefore the text has been corrected accordingly (page 18 lines 1 and 10 from bottom). Moreover, we added a sentence explaining that aS26 is at least present in Crenarchaeota/Lokiarchaeota and the two references are cited (see page 19, lines 2-4).

8) Title/keywords list.

The title maybe too specific and does not fully reflect ALL the new insights of this study.

Maybe some additional specific keywords should also be incorporated to ensure appropriate visibility of the work

The title has been changed into: The 30S:mRNA:aIF1A:aIF2:GTP:Met-tRNA^{iMet} cryo-EM structure reveals archaeal features and gives insights into evolution of translation initiation. We also have edited the keywords.

REVIEWERS' COMMENTS:

Reviewer #1 (Remarks to the Author):

In the revised version of the manuscript, the authors have thoroughly and exhaustively addressed all points raised during the first round of peer review. In my opinion, the manuscript is of an excellent quality and should be published in its current form in *Communications Biology*.

Reviewer #2 (Remarks to the Author):

In the revised version of Coureux et al. "The 30S:mRNA:aIF1A:aIF2:GTP:Met-tRNA^{iMet} cryo-EM structure reveals archaeal features and gives insights into evolution of translation initiation" authors adequately addressed all of my concerns. I recommend this manuscript for publication in *Communications Biology*.

Reviewer #3 (Remarks to the Author):

The authors have satisfactorily replied to all my comments. The work is very interesting and will certainly become a key reference in the field.

I have few additional suggestions that may facilitate the overall reading of the manuscript.

Results: section "Identification of archaeal specificities of Pa-30S"

1) First occurrence of nucleotide numbering: at several instance position of rRNA modifications are provided, please specify in the text if it is *E. coli* or *P. abyssi* numbering (I believe it is *Pab* throughout but that should be clearly mentioned).

2) End of this result section "Taking into account the distribution of N4-acetylations throughout the 16S rRNA and the non-systematic modification of all CCG sequences, small RNA guides would likely assist Nat10/Kre33."

This hypothesis is not understandable without additional explanations. The idea of small RNA guides-dependent targeting should be introduced based on what we know about Nat10/Kre33-dependent modifications.

3) Ribosomal protein nomenclature:

There is some potential for confusion here.

I am not sure whether the authors are following the current proposed (Ban et al 2014) r-proteins nomenclature or not? If so, this reference should be added (unless I missed it).

In the Ban et al nomenclature the prefix ae is avoided, and e is preferred. Even if it is, in my opinion, a poor choice the author may stick to this if using it. For example, eS28 is mentioned as aeS28 or eS17 as eaS17 (note the typo as well).

For the case of eS26, I would avoid using aS26 as it could be confusing (it is not archaeal specific), maybe prefer using the following wording: "archaeal eS26 or archaeal homologue of eS26 is absent in euryarchaeota", same when it comes to eL41.

aeS3 is not used in the Ban et al nomenclature.

In any case, the authors should clarify the r-proteins nomenclature used across the manuscript and modify accordingly (text and figures).

4) Supplementary Data: Figure S1

The sequence alignment for aS21 was not visible in the pdf version of the Supplementary data. Check that it appears correctly in the final/accepted version.

Point-by-point responses to the reviews COMMSBIO-19-1340-A

We thank the reviewers for their positive and kind comments on our work. We are grateful for their meticulous work along the reviewing process that much improved the quality of our manuscript.

We provide below our point-by-point responses to the comments of reviewer 3.

REVIEWERS' COMMENTS:

Reviewer #1 (Remarks to the Author):

In the revised version of the manuscript, the authors have thoroughly and exhaustively addressed all points raised during the first round of peer review. In my opinion, the manuscript is of an excellent quality and should be published in its current form in *Communications Biology*.

Reviewer #2 (Remarks to the Author):

In the revised version of Coureux et al. "The 30S:mRNA:aIF1A:aIF2:GTP:Met-tRNA^{iMet} cryo-EM structure reveals archaeal features and gives insights into evolution of translation initiation" authors adequately addressed all of my concerns. I recommend this manuscript for publication in *Communications Biology*.

Reviewer #3 (Remarks to the Author):

The authors have satisfactorily replied to all my comments. The work is very interesting and will certainly become a key reference in the field.

I have few additional suggestions that may facilitate the overall reading of the manuscript.

Results: section "Identification of archaeal specificities of Pa-30S"

1) First occurrence of nucleotide numbering: at several instance position of rRNA modifications are provided, please specify in the text if it is *E. coli* or *P. abyssi* numbering (I believe it is *Pab* throughout but that should be clearly mentioned).

We now clearly mention that the *P. abyssi* numbering is used, unless otherwise stated (page 8).

2) End of this result section "Taking into account the distribution of N4-acetylations throughout the 16S rRNA and the non-systematic modification of all CCG sequences, small RNA guides would likely assist Nat10/Kre33."

This hypothesis is not understandable without additional explanations. The idea of small RNA guides-dependent targeting should be introduced based on what we know about Nat10/Kre33-dependent modifications.

We agree with the reviewer that the sentence was unclear. It was corrected accordingly (page 9).

3) Ribosomal protein nomenclature:

There is some potential for confusion here.

I am not sure whether the authors are following the current proposed (Ban et al 2014) r-proteins nomenclature or not? If so, this reference should be added (unless I missed it). In the Ban et al nomenclature the prefix ae is avoided, and e is preferred. Even if it is, in my opinion, a poor choice the author may stick to this if using it. For example, eS28 is mentioned as aeS28 or eS17 as eaS17 (note the typo as well).

For the case of eS26, I would avoid using aS26 as it could be confusing (it is not archaeal specific), maybe prefer using the following wording: "archaeal eS26 or archaeal homologue of eS26 is absent in euryarchaeota", same when it comes to eL41.

aeS3 is not used in the Ban et al nomenclature.

In any case, the authors should clarify the r-proteins nomenclature used across the manuscript and modify accordingly (text and figures).

We now clearly mention that the Ban et al. nomenclature is used and the corresponding paper is now cited. The proteins were re-named throughout the manuscript, including figures and supplementary material, according to this nomenclature. We thank the reviewer for this suggestion.

4) Supplementary Data: Figure S1

The sequence alignment for aS21 was not visible in the pdf version of the Supplementary data. Check that it appears correctly in the final/accepted version.

We apologize for the unclear explanation concerning aS21 sequence alignment. It is now clearly stated in a "Description of additional supplementary items" file that the alignment is provided as a separate file in the FASTA format.